# Global decline in net primary production underestimated by climate models
Thomas J. Ryan-Keogh [1] ✉, Alessandro Tagliabue [2] & Sandy J. Thomalla [1,3]

Marine net primary production supports critical ecosystem services and the carbon cycle. However, the lack of consensus in the direction and magnitude of projected change in net primary production from models undermines efforts to assess climate impacts on marine ecosystems with confidence. Here we use contemporary remote sensing net primary production trends (1998–2023) from six remote sensing algorithms to discriminate amongst fifteen divergent model projections. A model ranking scheme, based on the similarity of linear responses of net primary production to changes in sea surface temperature, chlorophyll-*a* and the mixed layer, finds that future declines in net primary production are more likely than presently predicted. Even the best ranking models still underestimate the sensitivity of declines in net primary production to ocean warming, suggesting shortcomings remain. Reproducing this greater temperature sensitivity may lead to even larger declines in future net primary production than presently considered for impact assessment.

Marine net primary production (NPP) by phytoplankton sustains biodiversity and is essential to global ocean ecosystems, but its future is uncertain[1]. Despite its importance for the assessment of climate change impacts on the marine system[2], there is currently a lack of consensus regarding the sign of predicted change in NPP on regional and global scales under high emissions scenarios of the sixth coupled model intercomparison project (CMIP6)[3,4]. This leads to divergent global trends over the contemporary and future periods when comparing individual models (Fig. 1a) and a negligible projected change with a large standard deviation ($-0.76 \pm 3.44$ Pg C year$^{-1}$ by 2100[3]; Fig. 1b) when these divergent trends are averaged for the multi-model ensemble mean. Importantly, this NPP trend uncertainty has increased by more than 50% since the previous IPCC assessment cycle[3]. Furthermore, upper trophic level models that assess future responses of fisheries typically subsample NPP projections from the 'high' and 'low' extremes of available projections[5,6]. This highlights the urgent need to interrogate Earth system models and discriminate amongst them in order to deliver increased confidence in projections of NPP in response to climate change.

Emergent relationships between changes in remote sensing estimates of NPP and concomitant changes in ocean environmental conditions over the contemporary period can provide global constraints for Earth system models. Emergent constraints have been used to refine assessments of NPP trends in the tropical Pacific[7,8] and ocean carbon uptake in the Arctic and Southern Ocean[9,10] based on single parameter assessments but have yet to be exploited on a global scale or with multi-parameter relationships. Trends in

marine NPP estimated from remote sensing however also vary considerably depending on the time period, algorithm implemented, and data product being used[11–14]. Some of the sensitivities to time period and data product are addressed by the generation of a coherent multi-sensor satellite record spanning 1998–2023 that merges all available single-sensor satellite missions with substantially reduced inter-sensor biases[15]. Nonetheless, intrinsic differences in remote sensing trends are still apparent in the range of algorithms available for quantifying NPP rates. Here we focus on six algorithms including: (1) the 'vertically generalised production model's (Eppley-VGPM[16] and Behrenfeld-VGPM[17]), which define phytoplankton growth as a function of chlorophyll-a, light and temperature, the difference being that Eppley-VGPM is an exponential function of temperature, while Behrenfeld-VGPM is a 4th order polynomial; (2) the 'carbon-based production models (Behrenfeld-CbPM[18] and Westberry-CbPM[19]), which incorporate particulate backscatter as a proxy for phytoplankton carbon but differ in that Westberry-CbPM is both depth and wavelength resolved whilst Behrenfeld-CbPM is not; (3) the 'absorption-based production model' (Lee-AbPM[20]), which defines NPP as a function of phytoplankton absorption rather than chlorophyll; and (4) the 'carbon, absorption, and fluorescence euphotic' resolving model (Silsbe-CAFE[21]), which integrates the learning from all the above algorithms to define NPP as a function of energy absorption and efficiency (for more details please see Methods).

In this work, we rank fifteen CMIP6 Earth system models according to their ability to capture the emergent contemporary relationships observed between NPP and environmental variables (sea surface temperature,

[1]Southern Ocean Carbon-Climate Observatory, CSIR, Cape Town, South Africa. [2]Department of Earth, Ocean and Ecological Sciences, School of Environmental Sciences, University of Liverpool, Liverpool, UK. [3]Marine and Antarctic Research Centre for Innovation and Sustainability, Department of Oceanography, University of Cape Town, Cape Town, South Africa. ✉e-mail: tryankeogh@csir.co.za

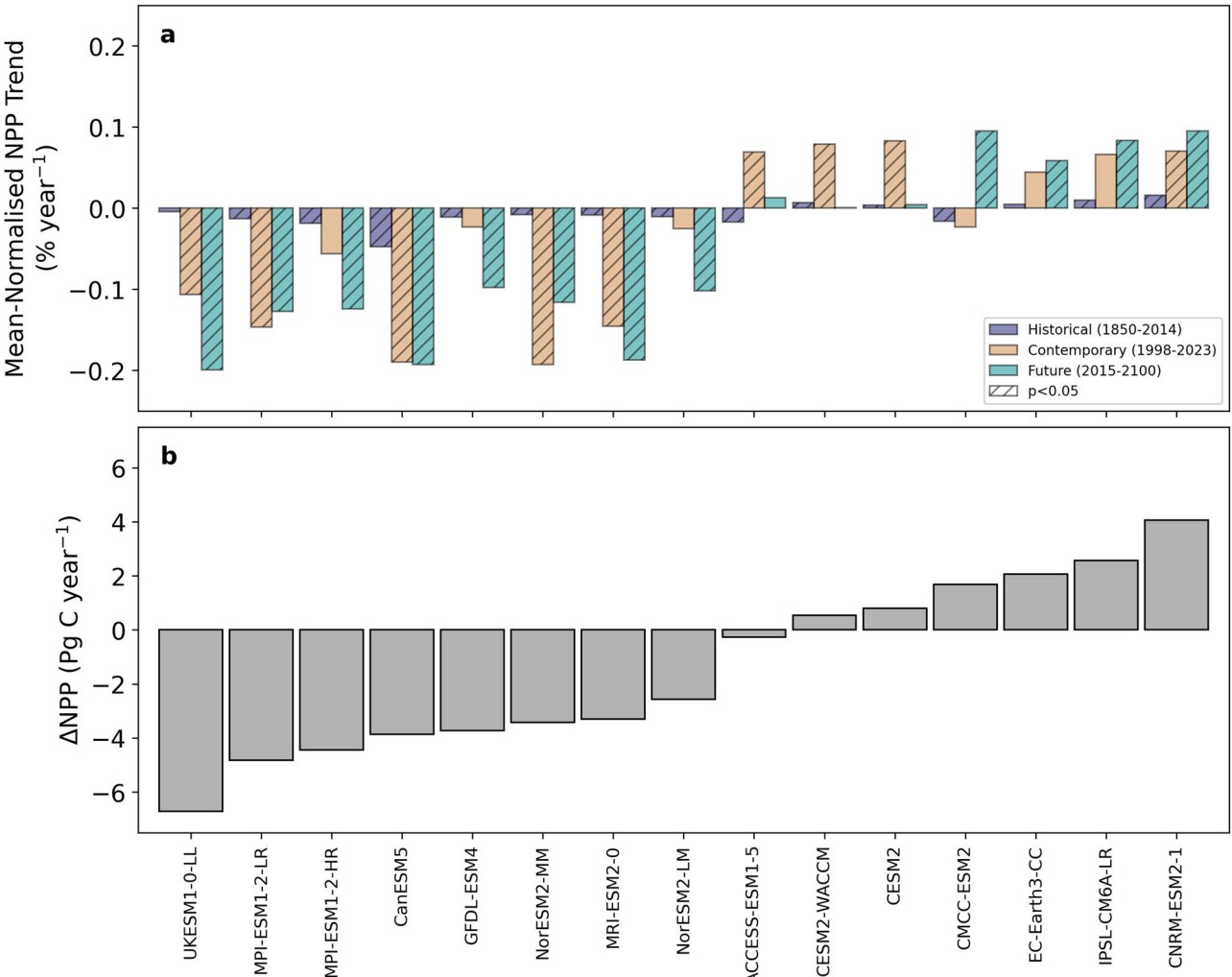

**Fig. 1 | Variability of net primary production trends from CMIP6 Earth system models. a** Area-weighted mean-normalised net primary production (NPP) annual trends (% year$^{-1}$) calculated using ordinary least squares for the historical (1850–2014), contemporary (1998–2023) and future (2015–2100) periods for the CMIP6 Earth system model ensemble. **b** Area-weighted ΔNPP (Pg C year$^{-1}$), calculated as the difference between the end of the historical period (1995–2014) and the end of the century (2081–2100), for each of the Earth system models in the CMIP6 ensemble. Both panels are sorted by ΔNPP from low to high values.

chlorophyll-*a* and the mixed layer depth) in the 6 remote sensing algorithms. Five of the remote sensing algorithms (Behrenfeld-VGPM, Behrenfeld-CbPM, Westberry-CbPM, Lee-AbPM and Silsbe-CAFE) concur that climate models projecting NPP declines rank higher, whilst the remaining algorithm (Eppley-VGPM) rank models that project positive NPP trends higher. These results suggest that future NPP decline is more likely than not, and this decline is currently underestimated by even the best ranked CMIP6 models, which predict the most intense NPP declines.

## Results and discussion
### Contemporary global trends in remote sensing NPP and their drivers

Remote sensing derived NPP trends are potentially susceptible to extreme climate events that align with either the start or end of the time series, e.g., El Niño-Southern Oscillation, which could act to increase or decrease estimated rates of change. This is more likely to impact the VGPM algorithms, which parameterise NPP as a function of temperature. To account for this, we performed a Monte Carlo jackknife resampling of ~80% of the time series (i.e., 20 of the 26 years - representing 7 different assessments) for all the analyses presented here. NPP trends from remote sensing are predominantly negative across the algorithms, apart from the two VGPM approaches which show positive trends, most notably at higher latitudes (>40°) (Fig. 2a, b). We note however that when globally averaged, the

VGPM trends have standard deviations that are larger than the mean, which reduces confidence in the NPP trends that these two algorithms produce (−0.004 ± 0.10% year$^{-1}$ & 0.04 ± 0.07% year$^{-1}$ respectively; Fig. 2a, b). The remaining four algorithms all produce globally averaged trends of declining NPP that range from −0.27% year$^{-1}$ to −1.45% year$^{-1}$ (Fig. 2c–f). The predominantly negative global trends from these four algorithms are more robust than the VGPM algorithms given that the standard deviations are always lower than the mean (0.14–0.23% year$^{-1}$). When trends across all algorithms are grouped into ocean provinces (Supplementary Fig. S1a[22]), the typical decline in remote sensing derived NPP across biomes and algorithms is emphasised (Supplementary Fig. S1b), apart from some high latitude regions in the Arctic and the Southern Ocean, where only the VGPM algorithms suggest an increase in NPP.

Trends in NPP occur in response to concomitant modifications of the ocean environment that span 'bottom up' factors like resource limitation to 'top down' controls such as grazing. Contemporary evidence already exists for climate-driven adjustments in sea surface temperatures[23] and the nutrient and light environment from altered stratification and mixed layer depths[24]. Such adjustments will impact phytoplankton physiology and their photosynthetic capacity, which is reflected in global trends in chlorophyll-*a*[25]. However, we lack insight into the relative roles played by different factors that shape contemporary trends in NPP across the different algorithms. To statistically assess what

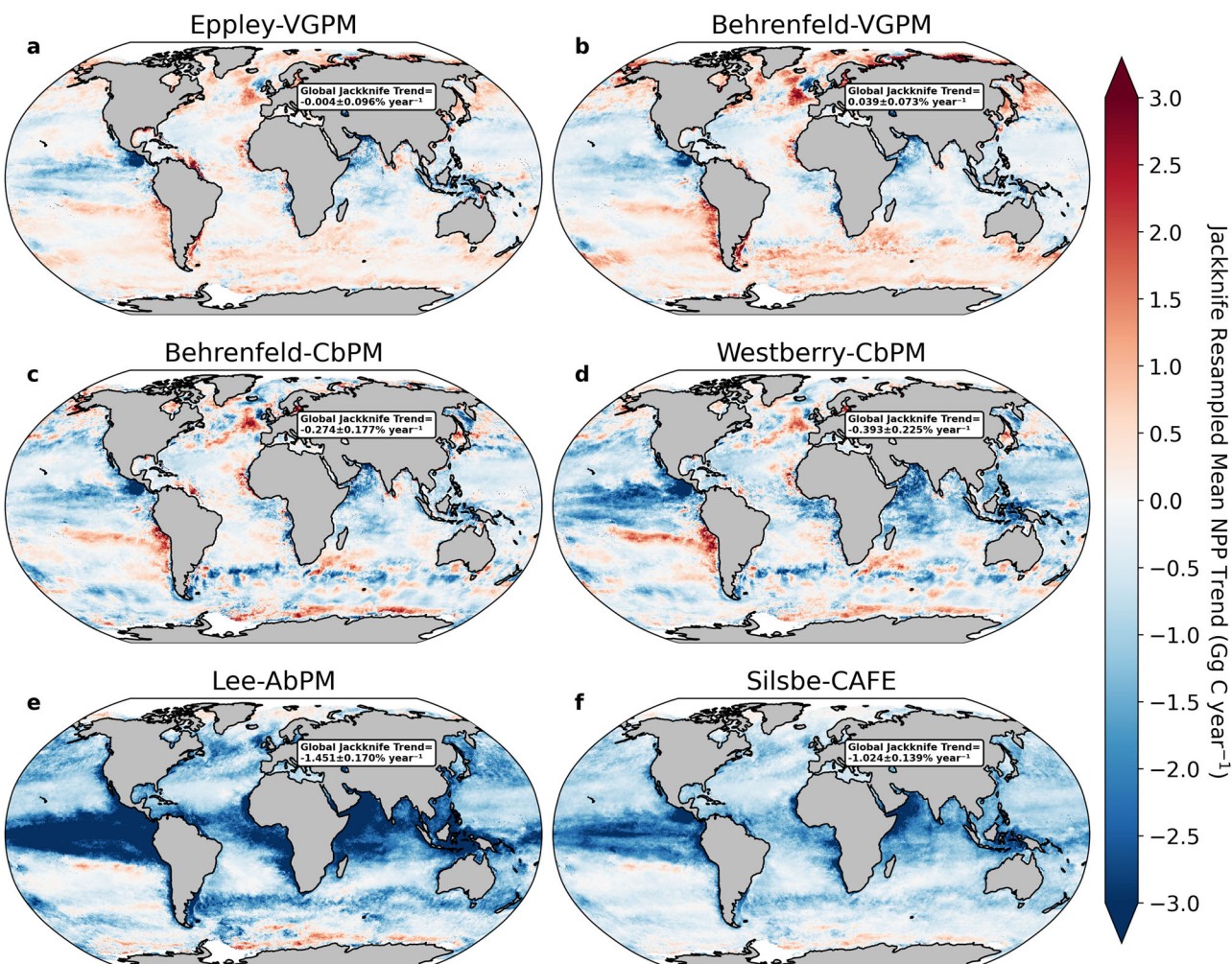

**Fig. 2 | Comparing jackknife resampled trends of net primary production from different remote sensing algorithms.** Global distribution of the mean jackknife resampled trends (1998–2023) of annual mean net primary production (NPP; Gg C year⁻¹) from **a** Eppley-VGPM, **b** Behrenfeld-VGPM, **c** Behrenfeld-CbPM, **d** Westberry-CbPM, **e** Lee-AbPM and **f** Silsbe-CAFE. Inset text reports the area weighted mean ±1σ jackknife resampled (80% of the 26-year time series) NPP trend (% year⁻¹) as displayed in Supplementary Fig. S1b.

drives local trends in NPP, we use multiple linear regressions (MLR) that account for unequal variance and autocorrelation. We used MLRs to link contemporary trends in NPP to a suite of environmental and biological drivers across all algorithms and jackknife trend assessments (see Methods). These drivers are trends in annual mean sea surface temperature (SST; where warming increases phytoplankton metabolic rates and may retard nutrient supply due to greater ocean stratification), annual mean chlorophyll-*a* concentration (CHL; which reflects phytoplankton biomass and physiology), and annual mean mixed layer depth (MLD; which impacts adjustments in both light and nutrient supply). These variables are all available from satellite remote sensing or compiled data products and importantly are also accessible as standard outputs from CMIP6 Earth system models. A sensitivity analysis of the entire time series showed that including all three drivers into the MLR analysis led to an increase in the global coverage of significant ($p < 0.05$) regressions and a higher median adjusted $R^2$ across all six remote sensing algorithms, as opposed to using either SST or CHL alone, or a combination of SST and CHL as predictors (Supplementary Fig. S2). The MLR applied to the Eppley-VGPM and Behrenfeld-VGPM NPP trends had the highest globally averaged mean adjusted $R^2$ values (Supplementary Fig. S3a, b), decreasing (most notably at high latitudes) for Behrenfeld-CbPM and Westberry-CbPM (Supplementary Fig. S3c, d), whilst Lee-AbPM and Silsbe-CAFE had a more even global spread and lower global

averages (Supplementary Fig. S3e, f). The higher global mean $R^2$ values for the VGPM algorithms is perhaps not surprising as the MLR is constructed using two of the three algorithm inputs, SST and CHL, with photosynthetically active radiation the remaining input variable.

The MLR coefficients associated with each driver show a reduction in amplitude, roughly halving in strength from SST to CHL and again from CHL to MLD (Fig. 3a–c). This indicates that trends in SST and CHL are the most important predictors of trends in NPP, whilst MLD plays only a minor role. Zonally averaged, the distribution of MLR coefficients for SST reveals regional coherence at high latitudes across all algorithms (Fig. 3a), whereas at mid to low latitudes the Eppley-VGPM algorithm behaves anomalously by displaying a positive relationship between SST and NPP, while all other algorithms display negative SST coefficients (accentuated at equatorial latitudes). Negative coefficients are indicative of declining NPP as the surface ocean warms, potentially reflecting the role of nutrient limitation from a reduced surface reservoir as stratification intensifies. The atypical positive coefficients from Eppley-VGPM on the other hand may reflect a metabolic response that favours increased growth rates under warmer conditions[26]. The CHL coefficient is consistently positive across all six remote sensing NPP algorithms, corresponding to the regulation of NPP by phytoplankton standing stocks (Fig. 3b). Whilst both the VGPM and Silsbe-CAFE algorithms display minimal latitudinal gradients in the magnitude of the CHL coefficient, the three other algorithms display higher equatorward

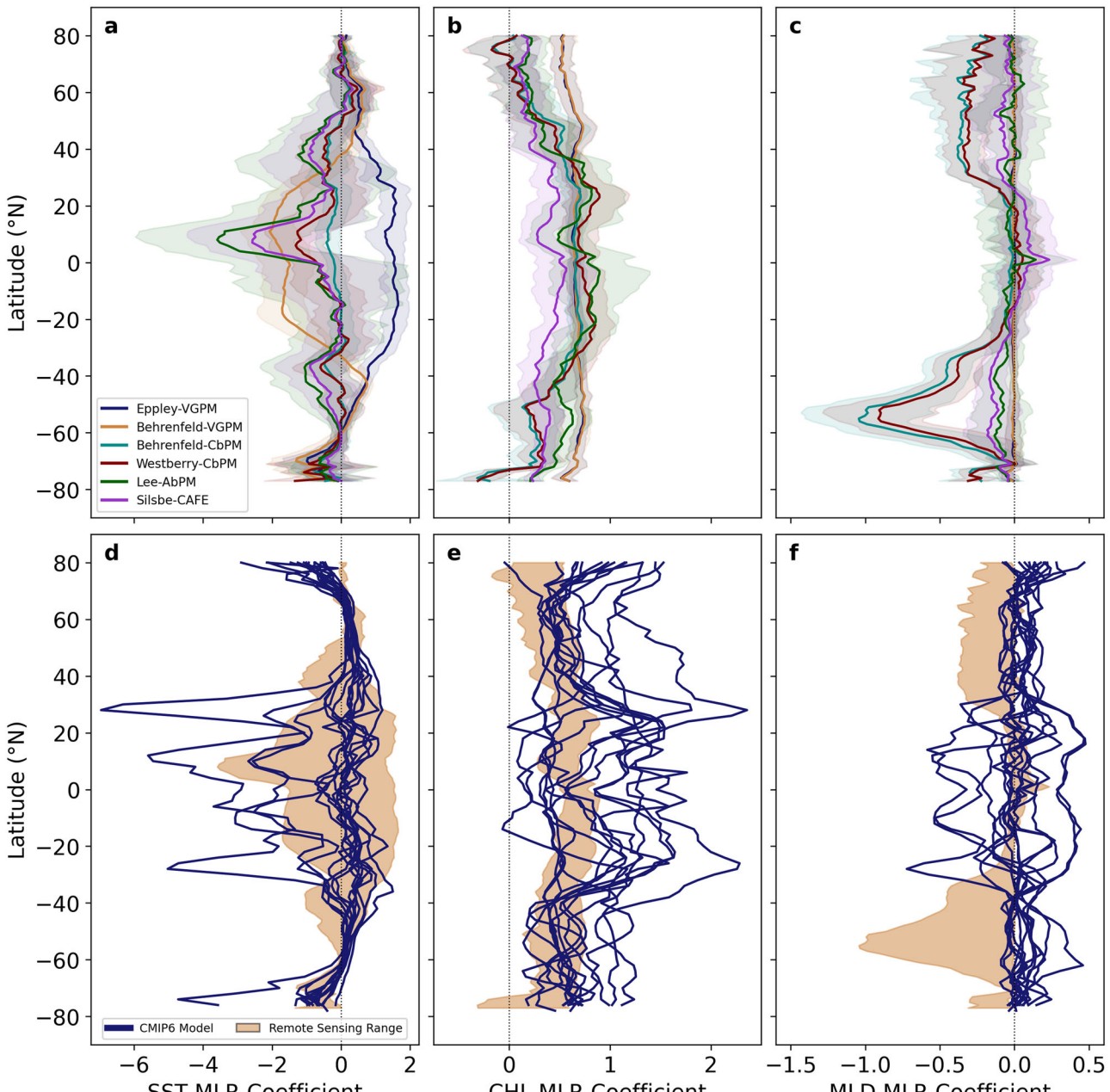

**Fig. 3 | Comparing the spatial variability of the dominant multiple linear regression coefficients between remote sensing and Earth system models.** Zonal averages ± standard deviations of the multiple linear regression coefficients for **a**, **d** sea surface temperature (SST), **b**, **e** chlorophyll-a concentrations (CHL) and **c**, **f** mixed layer depth (MLD) for the **a**–**c** Eppley-VGPM, Behrenfeld-VGPM, Behrenfeld-CbPM, Westberry-CbPM, Lee-AbPM and Silsbe-CAFE NPP algorithms and **d**–**f** the ensemble of CMIP6 Earth system models. Only pixels/grid points where the multiple linear regression analysis was significant are included in the zonal averages. The shaded region in panels **d**–**f** represents the range of coefficients as estimated from the remote sensing algorithms zonal averages (panels **a**–**c**).

coefficients that decrease towards the poles. MLD (Fig. 3c) shows a broadly similar pattern across all algorithms with regression coefficients at low latitudes that are close to zero, while negative coefficients are more prominent at higher latitudes (exaggerated in the two CbPM algorithms relative to the others). Consistent across all algorithms the spatial standard deviations (Fig. 3a–c) were always substantially greater than the standard deviations of the jackknife assessments (Supplementary Fig. S4), confirming that regional variability plays a more significant role in the proposed ranking scheme.

### Assessing Earth system model projections of NPP

Using an ensemble of fifteen Earth system models from CMIP6 we evaluate modelled trends in NPP (Fig. 1) in relation to the same set of drivers used in

the remote sensing analysis to develop a 'process based' model ranking scheme. MLR analysis between modelled NPP and the three associated drivers (SST, CHL and MLD) were repeated globally for each Earth system model over the historical period (1850–2014) but without jackknife resampling. This approach is similar to previous model assessment exercises that focussed on a single driver variable for a specific region (e.g. refs. 7,9,10), whereas our assessment is global and incorporates three driver variables. Overall, the MLR analysis applied to the Earth system model data produced a similar range in skill across models as for the remote sensing algorithms (Supplementary Fig. S5).

Both the magnitudes and spatial distribution of the MLR coefficients across SST, CHL and MLD for each Earth system model reveal stark differences, relative to the remote sensing assessment (Fig. 3d–f). However, the

general decline in their relative contribution to NPP trends from SST to CHL and lastly MLD largely remains, albeit to a lesser extent than the remote sensing algorithms. Whilst all CMIP6 models appear to place similar weights on SST at high latitudes, that are within the range of what was found for the six remote sensing NPP algorithms, at mid to low latitudes they vary markedly and are generally underestimated relative to remote sensing. This implies that CMIP6 NPP trends in the tropics and subtropics are not as sensitive to warming as remote sensing NPP trends to changes in SST (Fig. 3d). In addition, the CMIP6 models also tend to reflect a stronger positive influence of CHL on NPP, with coefficients that are almost double those found in the remote sensing algorithms, which suggests that some models may be oversensitive to CHL trends, most notably at high latitudes (Fig. 3e). MLD coefficients show a large discrepancy in their zonal distribution relative to remote sensing (Fig. 3f) with coefficients that are broadly positive and more similar to each other at high latitudes but larger and more diverse (spanning both positive and negative coefficients) at lower latitudes.

To constrain NPP projections we develop a regionally informed model ranking scheme by comparing the distribution of the MLR coefficients from each model with those from the jackknife analysis of each remote sensing algorithm. Models with a more similar distribution in their respective MLR coefficients (relative to a specific remote sensing algorithm) will rank higher than models with a very different distribution in their MLR coefficient values. This ranking is based on the dimensionless Earth mover's distance (EMD) metric[27], which quantifies the effort required to transform the distribution of the Earth system model MLR coefficients to match those obtained from each of the six remote sensing NPP algorithms. A low EMD value indicates that the Earth system model MLR coefficients closely match, i.e. are in good agreement, to those of the remote sensing algorithms. EMD metrics were calculated per jackknife assessment and per biome[22], with MLR coefficients restricted using the interquartile range fence test (see Methods) to remove extreme outliers in each biome. The EMD metrics, for SST, CHL and MLD were then weighted according to each biome's proportion of the globe and finally globally averaged to generate a single EMD value per MLR coefficient per model for each of the six remote sensing algorithm jackknife assessments (Supplementary Fig. S6). EMD metrics were generally higher for SST, followed by CHL and then MLD. The higher EMD values for SST imply a large disparity in the weighting of the MLR coefficients between remote sensing and models, which implies an inaccurate representation of the relationship between warming and NPP in the models.

Next, we average the EMD values across all three variables, SST, CHL and MLD, to generate a single EMD mean and standard deviation for each Earth system model per remote sensing algorithm jackknife assessment. For each remote sensing algorithm, we then rank the CMIP6 models using Z-scores that incorporate both the EMD mean and standard deviation. The Z-score is defined as the distance of a value to the group mean, such that high Z-scores indicate values that are atypical and much larger than the mean and vice versa. A low Z score thus indicates that the NPP driver relationship in the Earth system model more closely matched that of the remote sensing algorithm. We then combine both Z-scores (from the EMD mean and standard deviation) using equal weighting (i.e. we averaged the Z-scores), before sorting the combined Z-scores from smallest to largest to rank each Earth system model's relative performance (Fig. 4). However, since each remote sensing algorithm is made up of 7 jackknife assessments, each of the 15 Earth system models is independently ranked 7 times (for each remote sensing algorithm) (Supplementary Fig. S7). Model ranking results are presented with respect to the mean ΔNPP (Fig. 4), which is averaged across the 7 model rankings. When the same model is ranked in the same position for all 7 of the jackknife assessments there is no standard deviation in ΔNPP, whereas a standard deviation reflects instances where ΔNPP was averaged across more than one model with the same ranking. Five algorithms concur that climate models projecting greater NPP declines rank higher, whilst the

remaining algorithm (Eppley-VGPM) ranks models that project slightly positive NPP trends higher (Fig. 4).

## Assessing the merits of the different remote sensing algorithms

The ranking scheme results in a split between five algorithms that rank models that have negative NPP projections higher and one (Eppley-VGPM) that favours models with positive NPP projections. Focussing first on the two VGPM algorithms, we find that trends in NPP from Eppley-VGPM and Behrenfeld-VGPM are primarily driven by SST and CHL (Supplementary Fig. S8a, b). The Eppley-VGPM algorithm parameterises phytoplankton growth using an exponential function of temperature, explaining why it ranks positive NPP projections higher (Fig. 4a). However, only the Behrenfeld-VGPM algorithm implements a penalty on growth when temperatures increase beyond a certain threshold, consistent with its favouring of models with negative NPP projections (Fig. 4b). The trends in NPP from the CbPM algorithms are predominantly driven by changes in CHL (Supplementary Fig. 8c, d), rather than changes in particulate backscatter, similar to previous studies[28]. Differences in the ranking between the two CbPM algorithms are likely because the Behrenfeld-CbPM algorithm does not account for changing light properties through the water column (Fig. 4b), whereas Westberry-CbPM does (Fig. 4c). Finally, NPP trends in the Lee-AbPM and Silsbe-CAFE algorithms, which more consistently rank negative projections highest (Fig. 4e, f), are driven by trends in phytoplankton absorption (Supplementary Fig. S8e, f). In so doing, when determining phytoplankton productivity, these algorithms respond to the efficiency of light absorption, rather than the absolute quantity of photosynthetic pigments, which makes them better suited to capturing NPP responses to environmental variability[29].

During Primary Production Algorithm Round Robin exercises[30–32] no single algorithm has been found to perform best at all times and locations. However, there is a general reduction in the root mean square difference between remote sensing NPP estimates and direct field measurements for the Lee-AbPM and Silsbe-CAFE algorithms (relative to the VGPM and CbPM algorithms), suggesting that they perform best overall (Supplementary Fig. S9[21,32–34]). Indeed, more recent studies that applied the Behrenfeld-VGPM, Westberry-CbPM and Lee-AbPM algorithms to OC-CCI data report similar findings where Lee-AbPM has the lowest RMSE[34]. In addition, the jackknife trend analysis we conducted on the time series (Supplementary Fig. S10) demonstrates that both the Eppley-VGPM and Behrenfeld-VGPM algorithms are strongly sensitive to the start or end dates of the time series (Supplementary Fig. S10a–d), with high coefficients of variation and even a switch in the dominant direction of NPP trends across the assessments. Although both CbPM algorithms had similarly high coefficients of variation across the globe (relative to the VGPM algorithms), they remain dominated by negative trends across all assessments, with some evidence of an increase in the magnitude of negative trends and the number of positive trends in response to a change in the start and end dates (Supplementary Fig. S10e–h). The Lee-AbPM and Silsbe-CAFE algorithms displayed the most robust response in NPP trends to the jackknife assessments, with much lower coefficients of variation and no tangible increase in the number of positive trends (with only a slight increase in the magnitude of negative trends, Supplementary Fig. S10i–l). Those areas of the globe that display relatively higher coefficients of variation (e.g. the Southern Ocean) thus represent regions with reduced confidence in the magnitude of the predominantly negative trends, but not in their direction. Overall, this indicates that there are larger uncertainties for global NPP trends from the VPGM and CbPM algorithms, relative to the trends estimated from Lee-AbPM and Silsbe-CAFE. Together these points of consideration around NPP algorithm validation and trend sensitivity to the jackknife assessments suggest that the Lee-AbPM and Silsbe-CAFE algorithms are the most robust and therefore best suited for the implementation of the model ranking scheme. Consequently, these results support a greater likelihood of global NPP declines into the future.

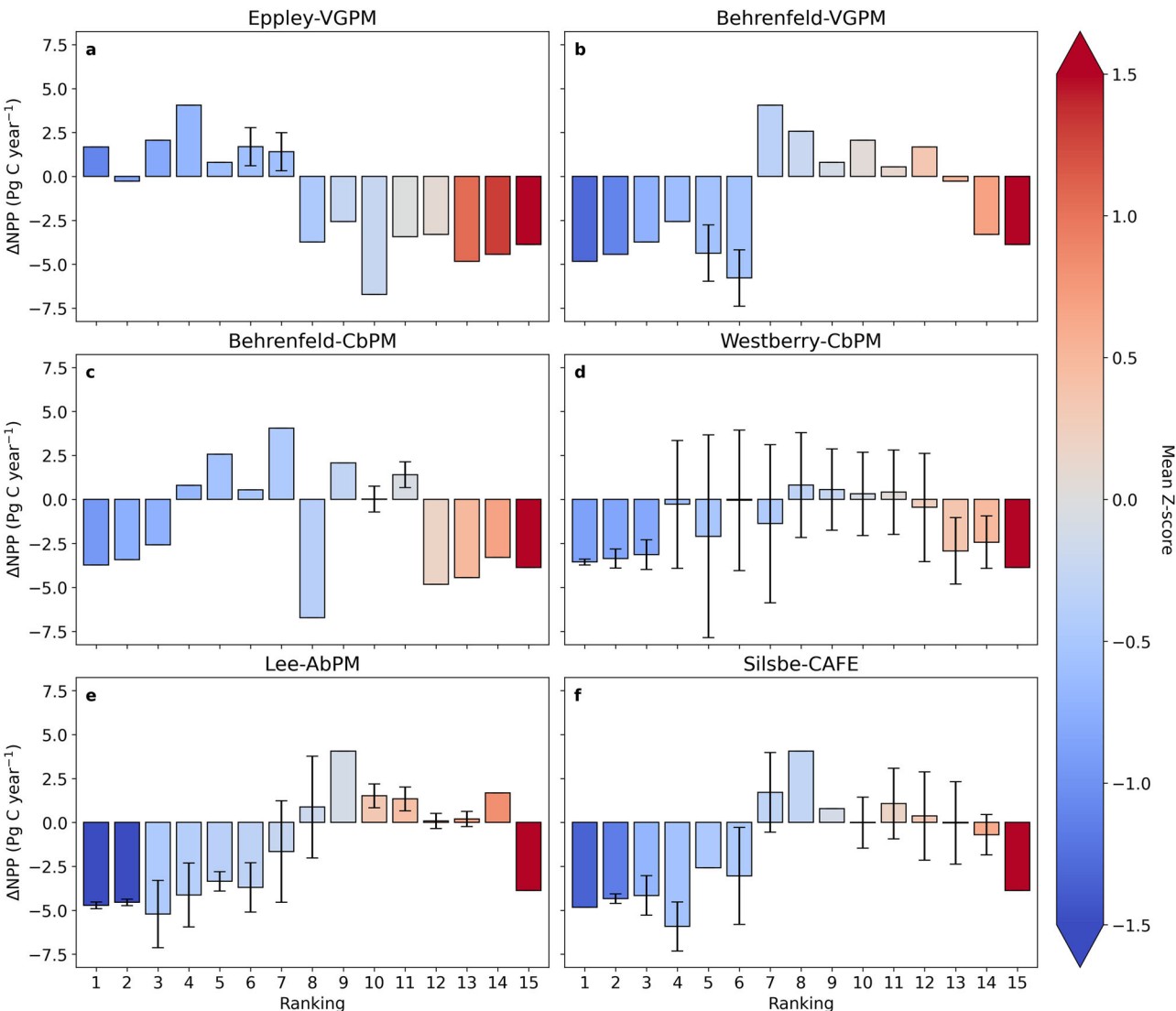

**Fig. 4 | Ranking Earth system models using Z-score assessments of the Earth mover's distance metric.** Bar plots of mean ± standard deviation jackknife resampled ranked Earth system model ΔNPP (Pg C year$^{-1}$) for **a** Eppley-VGPM, **b** Behrenfeld-VGPM, **c** Behrenfeld-CbPM, **d** Westberry-CbPM, **e** Lee-AbPM and **f** Silsbe-CAFE NPP algorithms. All bars are coloured by the mean Z-score across the jackknife resampling exercise. Please note that the absence of an errorbar is indicative of the same model being ranked in the same position for all 7 of the jackknife assessments.

## Are future declines in model NPP underestimated?

Using a multivariate model ranking scheme, based on the relationship between contemporary NPP and its associated drivers, we are able to discriminate amongst the Earth system model ensemble towards models that project future NPP declines. Despite advancements in model ranking through the application of our scheme, modelled trends of NPP over the contemporary period remain almost an order of magnitude too low (maximum range of ±0.2%; Fig. 1a) relative to remote sensing estimates (maximum range of −1.5%; Supplementary Fig. S1b). This reflects either too little sensitivity in the response of NPP to driver variability or incorrectly modelled trends in the drivers themselves. SST emerged as the most important driver of NPP, and the magnitude and direction of modelled contemporary trends in SST match the observational trends reasonably well, with a few model dependent regional exceptions (Supplementary Fig. S11a). The observed CHL trend on the other hand, which is the second most important driver of trends in NPP, is not well matched by Earth system models in magnitude, particularly at high latitudes, with some models also projecting regional differences in trend direction (Supplementary Fig. S11b). These differences in trend magnitude and oftentimes direction will translate into substantial differences in the weightings placed on SST and CHL for even the best

ranked Earth system models. For MLD, nearly all models project trends that are opposite in direction to the observations (Supplementary Fig. S11c) and tend to underestimate the role of MLD (Fig. 3f) on trends in NPP. That said, MLD plays a relatively small role in governing trends in NPP, such that inaccuracies in a given model's ability to capture trends in MLD will not strongly impact model performance. Accordingly, an improved reproduction of contemporary trends in NPP from Earth system models suggests NPP needs to become more sensitive to SST increases and less sensitive to CHL increases. Such a model would then become more sensitive to future levels of warming, suggesting that significant NPP declines may occur even under climate scenarios with strong degrees of mitigation that currently project stable NPP[4], with important ramifications for the ocean carbon cycle, marine ecosystem change and management.

Remote sensing is a powerful tool for understanding changes in ocean properties over the contemporary period, with multi-decadal records commonly used to assess and constrain Earth system models' ability to accurately represent spatial and temporal variability in ocean processes. The model ranking scheme applied here for NPP uses multiple metrics on a global scale to constrain trends and goes further than previous region-specific studies based on single driver metrics (e.g. ref. 7). Our model

ranking suggests a strong likelihood of negative future projections in NPP, particularly when based on the most robust remote sensing algorithms (Lee-AbPM and Silsbe-CAFE). Furthermore, the magnitude of the observed decline in NPP across those remote sensing algorithms is much larger than predicted change across even the best ranked Earth system models, suggesting a large underestimation of ongoing change in NPP across the global ocean. Future assessments should not only consider the uncertainties inherent to remote sensing algorithms (Supplementary Fig. S8), despite the complexity in deriving them[28] but should also expand on the Round Robin intercomparison exercises (Supplementary Fig. S9) as more in situ data becomes available. Furthermore, future model assessments should consider using additional parameters in combination with those proposed here, such as the resource limitation diagnostics in Earth system models (e.g. iron limitation, light limitation etc), which could be used to assess ongoing changes in the Southern Ocean[35] and the equatorial Pacific[36]. Such approaches would deliver greater confidence in the mechanistic representation of NPP in Earth system models necessary to project associated impacts on marine ecosystems and biogeochemical cycles.

## Methods

### Remote sensing net primary production calculations

Net primary production (NPP; mg C m$^{-2}$ d$^{-1}$) was calculated using the following algorithms, the 'vertically generalised production model's (Eppley-VGPM[16] and Behrenfeld-VGPM[17]; Eq. (1)), which relies on the relationship between chlorophyll $a$ and temperature derived growth rates; the 'carbon-based production model's (Behrenfeld-CbPM[18]; Eq. (2) and Westberry-CbPM[19]; Eq. (3)), which uses backscatter derived phytoplankton carbon as a biomass indicator and physiology derived as variability in the chlorophyll $a$ to carbon ratio; the 'absorption-based production model' (Lee-AbPM[20]; Eq. (4)), which does not make any assumptions on either chlorophyll $a$ or backscatter as biomass proxies but relies on absorption characteristics to infer phytoplankton photosynthetic efficiency; and the 'carbon, absorption, and fluorescence euphotic' resolving model (Silsbe-CAFE[21]; Eq. (5)), which derives NPP as a function of energy absorption and efficiency.

$$VGPM = Chl \times P_{opt}^{B} \times DL \times f(PAR) \times Z_{eu} \qquad (1)$$

where Chl is chlorophyll-a concentration (mg m$^{-3}$), $P_{opt}^{B}$ is a temperature (°C) based growth function (exponential for Eppley-VGPM and a 4th-order polynomial for Behrenfeld-VGPM), DL is day length (hours), f(PAR) uses PAR (mol photon m$^{-2}$ d$^{-1}$) the parameterized light term for the ratio of realised NPP to maximum potential NPP and $Z_{eu}$ is the depth of the euphotic zone (m).

$$Behrenfeld - CbPM = C_{ph}\left\{b_{bp}(\lambda443)\right\} \times \mu\left\{\mu_{max}, Chl: C_{ph}, I_g\right\} \times f(PAR) \times Z_{eu} \qquad (2)$$

where $C_{ph}$ is phytoplankton carbon (mg C m$^{-3}$) derived from an empirical relationship with particulate backscatter at 443 nm ($b_{bp}$ ($\lambda443$), m$^{-1}$), μ (μmax) is the growth rate, Chl:$C_{ph}$ is the chlorophyll-a to phytoplankton carbon ratio and $I_g$ is the growth irradiance term.

$$Westberry - CbPM = \int_{0}^{Z_{eu}} C_{ph}(z)\left\{b_{bp}(\lambda443)\right\} \times \mu(z)\left\{\mu_{max}, Chl: C_{ph}, I_g, Z_{NO3}\right\} \times dz \qquad (3)$$

where ZNO$_3$ is the depth of nitracline (m), defined as the depth at which nitrate + nitrite exceed 0.5 μM, and dz is the depth (m).

$$Lee - AbPM = f\left(a_{ph}(\lambda443) \times K_d(\lambda490) \times Z_{eu} \times PAR\right) \qquad (4)$$

where $a_{ph}(\lambda443)$ is phytoplankton specific absorption at 443 nm (m$^{-1}$), $K_d(\lambda490)$ is the light attenuation coefficient at 490 nm (m$^{-1}$) and PAR is the daily available photosynthetic radiation (mol photon m$^{-2}$ d$^{-1}$).

$$Silsbe - CAFE = Q_{PAR} \times \Phi_{\mu}^{max} \times \tanh(E_k/PAR(t, z, \lambda)) \qquad (5)$$

where $Q_{PAR}$ is energy absorption, $\Phi_{\mu}^{max}$ is the efficiency at which the absorbed energy is converted into carbon biomass and $E_k$ is the light saturation parameter. For more details on all equations please refer to their specific publications.

The algorithms were applied to ocean colour remote sensing data from the European Space Agency Ocean Colour Climate Change Initiative (OC-CCI) data product (8-day, version 6.0[15]) from 1998 to 2023, which was regridded to 25 km using bilinear interpolation. Photosynthetically active radiation (PAR; mol photons m$^{-2}$ d$^{-1}$) was taken from the merged GLOBColour product (http://globcolour.info) at 25 km 8-day resolution. For VGPM sea surface temperature (SST; °C) was taken from the Group for High Resolution Sea Surface Temperature (GHRSST; https://www.ghrsst.org/), which was regridded to 25 km as above. For CbPM and CAFE the mixed layer depth (MLD; m) was taken from the Hadley EN 4.2.2 gridded temperature and salinity profiles[37], which were first regridded to 25 km as above and resampled to 8-days, then converted to density using the Gibbs Seawater TEOS-10 python package and the MLD derived from a density criterion of 0.03 kg m$^{-3}$ and reference depth of 10 m[38]. Full explanation of the VGPM, CbPM and CAFE NPP calculations is provided by Ryan-Keogh et al.[39], with data publicly available[40]. For AbPM we used the OC-CCI $a_{ph}$ ($\lambda443$; m$^{-1}$) and $K_d$ ($\lambda490$) (m$^{-1}$), in combination with the GLOBColour PAR, with data publicly available here[41].

### Earth system model selection and download

CMIP6 data were obtained from the Earth System Grid Federation data server for the historical (1850–2014) and the high emission SSP5-8.5 (2015–2100) scenarios of the ACCESS (r1i1p1f1)[42], CESM2-WACCM (r1i1p1f1)[43], CESM2 (r4i1p1f1)[43], CMCC-ESM2 (r1i1p1f1)[44], CNRM-ESM2-1 (r1i1p1f2)[45], CanESM5 (r1i1p2f1)[46], EC-Earth3-CC (r1i1p1f1)[47], GFDL-ESM4 (r1i1p1f1)[48], IPSL-CM6A-LR (r1i1p1f1)[49], MPI-ESM1-2-HR (r1i1p1f1)[50], MPI-ESM1-2-LR (r1i1p1f1)[50], MRI-ESM2-0 (r1i2p1f1)[51], NorESM2-LM (r1i1p1f1)[52], NorESM2-MM (r1i1p1f1)[52] and UK-ESM1-0-LL (r1i1p1f2)[53] models. Data variables downloaded include depth integrated NPP ('intpp'; mol C m$^{-2}$ d$^{-1}$), SST ('tos'; °C), sea surface chlorophyll-$a$ concentration ('chlos'; kg m$^{-3}$) and MLD ('mlotst'; m). All data variables were regridded on a regular 1° × 1° grid using the bilinear interpolation of Climate Data Operators[54], and were resampled from a monthly resolution to annual means.

### Calculating trends

Trends of remote sensing annual mean NPP were calculated by first excluding any pixel whose time series had less than 50% of the data available. Before linear regressions were performed, the data were first tested for a normal distribution using the D'Agostino-Pearson test in the SciPy python package[55]. If the data were normally distributed, then linear regressions were performed using the Sci-Kit[56] Huber-Regressor, where ε, the parameter to control the amount of robustness (i.e. the number of outliers), was set to value of 1.35. This value is to ensure maximum robustness whilst maintaining 95% statistical efficiency[57]. If a pixel had less than 50% of the time series following outlier removal, then no further tests were performed. If the data were not normally distributed, then linear regressions were performed using the non-parametric Mann–Kendall Test[58]. This same method was applied to calculate trends in annual mean SST, annual mean chlorophyll-$a$ concentration (CHL) and annual mean MLD (MLD).

For spatially averaged biome annual mean NPP trends, either remote sensing or Earth system models, an ordinary least squares regression was applied to area weighted data normalised to the mean. Area weighting was determined as a function of latitude, such that remote sensing pixels/model grid points at higher latitudes are smaller than remote sensing pixels/model grid points at lower latitudes. ΔNPP (Pg C) for each Earth system model was calculated as the difference between the global averages from the historical

reference period (1995–2014) and the SSP5-8.5 scenario reference period (2081–2100). Please note that the remote sensing annual means of high latitudes may be potentially underestimated due to the presence of cloud cover preventing the retrieval of data.

## Multiple linear regression and earth mover's distances analyses

Annual means of NPP, SST, CHL and MLD were first jackknife resampled to 80% of the time series, representing 7 different possible assessments, and then mean-normalised, i.e. the resampled time series was divided by its mean. Multiple linear regressions (MLR) were then performed using the ordinary least squares function from the Statsmodel package[59] using a Heteroskedasticity and Autocorrelation Consistent covariance estimator, where the time lags for autocorrelation were calculated following the Newey & West[60] rule of thumb, defined in Eq. (6):

$$time\ lag = \left\lfloor 4 \times \left(\frac{T}{100}\right)^{\frac{2}{9}} \right\rfloor \tag{6}$$

where T is the length of the time series, which in this case is 26 years for remote sensing and 165 years for the Earth system models. No MLR was performed for a remote sensing pixel or model grid point if any variable was missing data from any year of the time series or if the variance for any of the drivers was ~0. MLR coefficients for each prospective driver were excluded from further analysis if either the remote sensing pixel or model grid point were not significant ($p > 0.05$). Comparisons between the observational data products and model coefficients were performed using the Earth mover's distance (EMD) metric[27], also known as the Wasserstein distance in mathematics[61] and Mallow's distance in statistics[62], defined here in Eq. (7):

$$l_1(u, v) = \int_{-\infty}^{+\infty} |U - V| \tag{7}$$

where $l_1$ is the first EMD, u and v are the respective distributions of the MLR coefficients from remote sensing and Earth system models and U and V are the respective cumulative distance functions of u and v. The MLR coefficient values for both the remote sensing and models were restricted using the interquartile range (IQR) fence test, IQR $\pm$ IQR$\times$3, to remove any extreme outliers within each ocean biome. The EMDs were calculated on a per biome basis using the biome classification of Fay & McKinley[22], with the EMD weighted by the biome's areal proportion (%) of the global ocean. The EMDs for SST, CHL and MLD were then averaged to generate an EMD mean and standard deviation per Earth system model. To rank the models the Z-score (Eq. (2)), also known as standard score, was calculated using Eq. (8):

$$z = \frac{x + \mu}{\sigma} \tag{8}$$

where x is either the model's EMD mean (or standard deviation), $\mu$ is the model ensemble mean of either the EMD mean (or standard deviation) and $\sigma$ is the model ensemble standard deviation of either the EMD mean (or standard deviation). The final Z-scores, determined from both the EMD mean and the EMD standard deviation, were then generated by combining with equal weighting (i.e. the Z-scores were averaged together).

## Data availability

Remote sensing NPP data for Behrenfeld-VGPM, Westberry-CbPM and Silsbe-CAFE are available at: https://doi.org/10.5281/zenodo.7849934. Remote sensing NPP data for Lee-AbPM are available at: https://doi.org/10.5281/zenodo.10014029. Ocean Colour Climate Change Initiative dataset, Version [6.0], European Space Agency, available online at http://www.esa-oceancolour-cci.org/. Ocean colour photosynthetically active radiation data retrieved from: https://www.globcolour.info/. Hadley EN4.2.2 temperature and salinity data were retrieved from: https://www.metoffice.gov.uk/hadobs/en4/download-en4-2-2.html. Sea surface temperatures were

retrieved from: https://www.ghrsst.org/. CMIP6 data were obtained from: https://esgf.llnl.gov/. All data used in this study are available at https://zenodo.org/records/14185537.

## Code availability

The code for reproducing all figures and data in the manuscript is available at https://github.com/tjryankeogh/global_npp_trends.

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

## Acknowledgements

We acknowledge the European Space Agency and the Ocean Colour Climate Change Initiative teams for making the remote sensing ocean colour data available. GlobColour data (http://globcolour.info) used in this study has been developed, validated, and distributed by ACRI-ST, France. We acknowledge the World Climate Research Programme's Working Group on Coupled Modelling, which is responsible for CMIP, and we thank the climate modelling groups (listed in the materials and methods of this work) for producing and making available their model output. We would like to thank Zhongping Lee for sharing the code for running the absorption-based production model. We would also like to thank Laurent Bopp and Chris Follett for advice. T.J.R.K. and S.J.T. were supported through the CSIR's Southern Ocean Carbon-Climate Observatory (SOCCO) Programme (http://socco.org.za/) funded by the Department of Science and Innovation (DSI/CON C3184/2023), the CSIR's Parliamentary Grant (0000005278) and the National Research Foundation (SANAP23042496681; MCR210429598142). A.T. received funding from the European Research Council (ERC) under the European Union's Horizon 2020 research and innovation program (grant agreement no. 724289) and the National Environment Research Council (NE/Y004531/1).

## Author contributions

Thomas Ryan-Keogh conceived the study, performed the analysis, developed the methodology, wrote the code, produced the figures, interpreted the results and took the lead in writing the manuscript. Alessandro Tagliabue conceived the study, developed the methodology, interpreted the results and provided inputs and refinement for writing the manuscript. Sandy Thomalla conceived the study, interpreted the results and provided inputs and refinement for writing the manuscript.

## Competing interests

The authors declare no competing interests.
