## [Peer review file · Communications Earth & Environment]

Global decline in net primary production underestimated by climate models

Corresponding Author: Dr Thomas Ryan-Keogh

Version 0:

Decision Letter:

Dear Dr Ryan-Keogh,

Your manuscript titled "Global decline in net primary production underestimated by climate models" has now been seen by 3 reviewers, whose comments are appended below. You will see that they find your work of some potential interest. However, they have raised quite substantial concerns that must be addressed. In light of these comments, we cannot accept the manuscript for publication in its current form, but would be interested in considering a revised version that fully addresses these serious concerns. In addition, please consider the following editorial thresholds: i) give clear evidence supporting the best two ranked models, and ii) explain in-deep the methods used, specially their reproducibility and the mismatch between modeling and field data.

We hope you will find the reviewers' comments useful as you decide how to proceed. Should additional work allow you to address these criticisms, we would be happy to look at a substantially revised manuscript. If you choose to take up this option, please either highlight all changes in the manuscript text file, or provide a list of the changes to the manuscript with your responses to the reviewers.

When resubmitting, please provide a point-by-point response to the reviewers' comments. Please submit your responses as a separate file, distinct from your cover letter where you can add responses to the Editors' comments that you do not want to be made available to the reviewers. Word files are preferred. We recommend that any figures, tables or graphs that are included in the response to reviewers are also included in the main article or Supplementary Information.

If the revision process takes significantly longer than three months, we will be happy to reconsider your paper at a later date, as long as nothing similar has been accepted for publication at Communications Earth & Environment or published elsewhere in the meantime.

Please use the following link to submit your revised manuscript, point-by-point response to the reviewers' comments with a list of your changes to the manuscript text (which should be in a separate document to any cover letter), a tracked-changes version of the manuscript (as a PDF file) and any completed checklist:

Link Redacted

Please do not hesitate to contact us if you have any questions or would like to discuss the required revisions further. Thank you for the opportunity to review your work.

Best regards,

Jose Luis Iriarte Machuca, PhD
Editorial Board Member
Communications Earth & Environment

Alice Drinkwater, PhD
Associate Editor
Communications Earth & Environment

EDITORIAL POLICIES AND FORMAT

If you decide to resubmit your paper, please ensure that your manuscript complies with our editorial policies and complete and upload the checklist below as a Related Manuscript file type with the revised article:

Editorial Policy Policy requirements
(Download the link to your computer as a PDF.)

- Behavioural and social science
- Ecological, evolutionary & environmental sciences
- Life sciences

<https://www.nature.com/documents/nr-reporting-summary.zip>

For your information, you can find some guidance regarding format requirements summarized on the following checklist: (<https://www.nature.com/documents/commsj-phys-style-formatting-checklist-article.pdf>) and formatting guide (<https://www.nature.com/documents/commsj-phys-style-formatting-guide-accept.pdf>).

REVIEWER COMMENTS:

Reviewer #1 (Remarks to the Author):

This is an important paper comparing marine Net Primary Production (NPP) from Earth System Models (ESM) and from satellite observations over the contemporary period (1998-2023). First the trends of NPP are presented, revealing that the observed NPP is declining and that the ESM are either not predicting or are underestimating this decrease. Six different remote sensing algorithms are used and compared, and fifteen ESMs are presented.

To evaluate which ESMs is modelling the best response to environmental drivers of NPP, multi linear regressions are computed between NPP and the drivers of NPP which are SST, chlorophyll-a and annual mean and max of MLD. This MLR is also done on observations for the six different algorithms and observed SST, chl-a and MLD. The resulting coefficients of regression of the fifteen ESMs are compared to the ones of the six remote sensing algorithms with an Earth movers' distance (EMD) metric, from which they extract a Z-score. A high Z-score indicates a poor relationship between NPP drivers.

The authors state that the ranking of the ESMs according to this Z-score confirms "a strong likelihood of negative future projections in NPP". The figure 4 and 5 are however not showing such a clear message. I am not convinced that there is a "strong" likelihood of negative future projections in ESMs' NPP according to this analysis. This paper would require to either tone down the message, or to bring "strong" elements to confirm this conclusion.

In more details :

1) The main result is that the "global decline in NPP is underestimated by climate models" over the contemporary period. This result is shown with the barplots of trends in % decade⁻¹, in figure 1 for the ESM and figure S2b for the remote sensing algorithms.

I am not convinced that the model ranking scheme is adding valuable insights to support this result, contrary to what is stated in the abstract L24 : "This scheme is able to sort models and in so doing reduce across-model variance with results suggesting that future declines in global NPP are more likely and currently underestimated."

"reduce across-model variance" is not that clear on the figure 4 and 5 :

- In figure 4, there is only three out of six remote sensing algorithm that are ranking the strong decline in NPP first, the three others are a bit random, Eppeley-VGPM and Behrenfeld-CbPM are even ranking the strong declines tenth or higher.

- In figure 5a, if we retain eight models out of fifteen for example, the DeltaNPP 1sigma increased for four of the remote sensing algorithm, but it should be decreasing if the ranking was selecting the models that are predicting a decrease of NPP.

- In figure 5, if we keep only the four best ranked models, two algorithms show no decrease in DeltaNPP 1sigma and three

algorithms show a positive DeltaNPP mean or equal to zero.

- Finally I do not see how the model ranking scheme is suggesting that the global NPP decline is underestimated, as stated in the abstract in the same sentence. Could the authors please explain further this statement?

2) How is it possible to compare the contemporary period of the ESMs and of the remote sensing data? I believe the ESMs are not forced with real data so the ESM years do not coincide with observed years?

Also, over which period is the MLR computed for the ESMs? Is it over 1998-2023 or over 1850-2100? I did not find this information in the paper, could the authors please explain further?

3) In figure S8 we see large variations in the Z-score. It would be interesting to display these variations against DeltaNPP, to add more information than the ranking. If the strongly decreasing ESMs have way smaller Z-scores than the other it would be a strong point for the paper. Could the authors display the Z-score in a scatterplot against DeltaNPP? Like a combination of Figure 4 and Figure S8. That would be 6 scatterplots on which we should discriminate the better scoring ESMs.

In line comments :

L245-246: "modelled trends of NPP over the contemporary period remain too low relative to remote sensing estimates" this statement is supported by figure 1b and figure S2b. It is the main result of this paper (title), could the authors add in figure 1 a column for observed NPP trend and DeltaNPP, next to the "ALL" column? To display 6 barplots (for the 6 algorithms) of trends and differences over the contemporary period that are comparable with the ESMs. Because for now we have to read supplementary figure S2b to assess the main result.

L70: Could the authors introduce briefly what differentiate Epley, Behrenfeld or Westberry algorithm? There is only a short description of the VGPM and CbPM (L61-62) but why is there 4 of them with different prefixes?

Figure 4 :

There is IPSL-CM6A-LR twice in the legend instead of MPI-ESM1-2-LR, could the authors double check that the colors correspond to the ESMs name?

Reviewer #2 (Remarks to the Author):

Summary

The manuscript first introduces that climate models from the CMIP6 often disagrees with the future trends of NPP (including increase or decrease) in business-as-usual scenario (SSP5-8.5) as well as the current NPP methods using satellites observations. The study claims that the two satellite methods (Lee-ABPM and Silsbe-CAFE) that show the highest NPP decline (Lee-ABPM and Silsbe-CAFE) are more accurate, and their response coefficients to SST, Chl-a, and MLD, can be compared to the same coefficients in climate models to rank the climate models that possess similar responses to the contemporary period from the satellites observations. This shifts to two main arguments that the best ranked models compared to Lee-ABPM and Silsbe-CAFE are the ones showing the highest NPP declines but that even the climate model NPP highest decline is underestimated when compared to the satellite observations. The manuscript suggests that its work can be used for improving confidence in climate models.

The conclusions are novel and interesting, and the manuscript is well written in most parts. However, the evidence should be strengthened, methodological choices better justified, and methods explained in more detail. In my opinion, the work is relevant to the field and is worthy of publication but only if the major and minor comments are addressed.

Major comments

The main claim of the study (Global decline in net primary production underestimated by climate models) strongly relies on demonstrating that Lee-ABPM and Silsbe-CAFE are the most accurate methods for estimating NPP by satellite observations because only those algorithms are showing the substantial decline not captured in the climate models. However, this demonstration evidence I found in the manuscript seems rather weak. One evidence is summarised in Table S1 and it seems to include data only on tropical regions. Given the different biomes considered in the global estimation, it is necessary to show better performance in more different regions. I also think it is necessary to give a plausible explanation on the reason both Lee-ABPM and Silsbe-CAFE performs better. Strengthening this part would better support the main claim of this study.

Furthermore, the methods section seems very hard to support the reproduction the study considering the information given. As I found the claim in this study quite important, the authors could expand and better describe the steps for reproducing the study. It could be beneficial for the authors, for example, to ask to a colleague to try to reproduce or understand all steps before resubmitting the manuscript if accepted. See the minor comments to more details.

Minor comments

"R" is for row or line

Summary paragraph

R20 Make it clear that the periods compared are not the same. Satellite trends are contemporary while Climate model trends are the difference from contemporary to 2100.

R23 Maybe “drive contemporary trends” is better explained as “based on the similarity of linear responses of NPP to SST, ETC, observed in the contemporary period”.

R38 Clarification is needed on how the areas are weighted and normalized. The variables are not shown in % per decade per m². Does it mean that larger areas have the same weight of small areas? Besides, why climate models and satellite estimations are not described as the same parameter most important for such study as Pg C per decade.

Main text

R68 Where do they best perform overall? The supplementary table S1 suggests that this performance is only assessed in tropical regions. Performance might differ depending on the site.

R71 “mechanistic”: to my understanding the relationships are statistical and not exactly mechanistic. For example, the SST influence is not clearly distinguished between influence on growth rates or relationship with MLD. Could it be better formulated?

Contemporary global trends in remote sensing NPP and their drivers

R101 How can decadal trends be sensitive to shorter oscillations as ENSO? Pending clarification on methodology. Is a moving 10-year window average? How was the edges problem dealt with it? A decadal time series of decadal NPP of satellite and models could better clarify that also.

R125 Are these regression over the decadal averages? Please, clarify it. If not, change to decadal averages as the study is focused on decadal trends.

R134 Two points should be considered here. First, the inclusion of new predictive variables can artificially increase the R². Showing the adjusted R² would better suit for the comparison. Second, clarify if and how the time auto-correlation of NPP was normalized for computing the p-value.

Assessing Earth system model projections of NPP

R169 “Over the historical period” Why use the historical period? As showed in Figure 1, the decadal trends vary among the periods: historical, contemporary, and future. Since this variability happens and the authors want to compare with the contemporary satellite observations, it seems more plausible to use the climate models under the contemporary period. Furthermore, it is not clear if the regression is on decadal averages. Considering the comparison with Figure S6, it would better suit to show the adjusted R² and how the time autocorrelation was normalized for estimating the p-value.

R194 Would be beneficial to explain what a low value and high value means.

R197 “10TH and 90TH percentile to remove extreme outliers”. The authors are removing 20% of the data, which does not sound as extreme outliers. Reduce it to 5% (2.5% and 97.5%) or give a plausible explanation of this wide range of removal.

R211 Please, better clarify the explanation of what is shown in figure S8. Are they the z-score mean and standard deviation of what distribution? I found this part difficult to understand.

R217 The dark grey in Figure 4 legend (Rank 2 in Lee-abpm) seems to be missing. Besides, one bar is too small that the colour is not visible, please, fix it.

Are future declines in NPP underestimated?

R250 Change “CHL” to the standard used in the text.

Material and Methods

Remote Sensing Net Primary Production

R311 Please, describe at least the simplified versions of each NPP equation applied in the satellite observations. Although some sort of normalised response is shown in Figure S1, it would be more much beneficial for the reader to assess the equations that the author employed. This could be easily done and very important as they are used to rank the climate models.

R320 Better describe the satellite products used. For example, clearly state where each variable used in the manuscript came from which product, the time frequency of each product, the uncertainty when available, and give the proper citation of these products (each website product often has a section describing how to cite their data when using them, follow those guidelines). It should also address some known limitations that might impact this work: i) given the optical observations are lacking in high latitudes during the winter, how the NPP was corrected given that there is no data during periods of very low NPP in this region; ii) the MLD thresholds used is 0.03 kg m⁻³, which may not be the same threshold used in some of the climate models (0.01 kg m⁻³), state this limitation and any foreseen potential impact.

Earth System Model Selection and Download

R341 Time frequency of the models?

Calculating Decadal Trends

R348 Clearly separate what is the method for the satellite trends and for the climate models trends.

R348 Describe in more details the method for estimating the trends. Is it a 10-year moving window average? How was the edge problem of the time series resolved? What is the period interval?

R348 What are the impact of removing the pixels in the high latitude regions on the estimations of global NPP? In satellite observations, it is often lacking data in these regions because of the long dark winter and cloudy conditions. See figure S5 and it is noticed the white areas in these parts. Given that high latitude regions are expected to have a substantial increase in NPP mainly driven by reduction of sea ice among others (some examples: <https://doi.org/10.1016/j.pocean.2015.05.002>, <https://doi.org/10.1038/s41467-020-20470-z>), how not including these positive anomalies could affect the global decadal trend estimate in this study?

R350 What normal distribution test? (e.g., Shapiro-wilk?)

R352 How can more than 50% of a distribution be the outliers? If they are a normal distribution, it would not be impossible to more than 50% of the data being outliers?

R352 What is ϵ used for and why 1.35?

R354 Could the different methods of estimating trends generate different slopes as an artificial outcome from the method given if it employed both in normal distributions?

Multiple Linear Regression and Earth Movers Distance Analysis

R364 Why annual means? The interannual modelled response would reflect the response in a decadal time scale? Is there any limitation on that?

R364 Why the variance was not normalized?

R365 The autocorrelation was checked but what was done about it? It is difficult to not find autocorrelation in annual means of the ocean given the oceans long memory, especially for SST. If auto correlation was found, how it was corrected in the regressions? Mention the auto correlation method instead of the tool, or both if necessary.

R366 Name the method for the regressions.

R369 Significance test and correction on autocorrelation if necessary?

R374 Correct the equation and describe below the variable I1.

R379 Why remove 20% of the data as outliers instead of 5%? I don't consider 5% as a standard value but 20% looks excessive and demands an explanation.

R387 Describe better this equation. For example, x is what value of the model, etc.

Reviewer #3 (Remarks to the Author):

COMMSENV-24-1567-T Review signed by Benoit Pasquier

manuscript title: Global decline in net primary production underestimated by climate models

The ocean's biological primary productivity sustains global marine ecosystems and is tightly linked to the global carbon cycle and climate.

Accurate projections of the ocean's productivity for the next century are thus critically important given the dramatic changes expected from our rapidly warming climate.

However, there is a major issue with the current state of the science.

Climate models disagree on the magnitude and even the sign of predictions of net primary production (NPP) over the next century.

A likely related issue is that remote-sensing estimates also disagree on the magnitude and sign of the NPP trend of the past few decades.

To tackle these issues, the authors compare and rank the CMIP6 climate models according to their ability to match historical NPP estimates from a suite of remote-sensing models.

An important feature of the climate-model rankings is that it is based on the sensitivities of NPP estimates to environmental variables.

The major claim of this paper, which is clearly laid out in the title, is that CMIP6 models underestimate the future NPP decline.

This claim is mainly supported by two arguments: (i) the rankings of climate models, of which those predicting strong future NPP declines tend to be ranked better, and (ii) the mismatch between the low SST sensitivities of climate models and the high SST sensitivities of remote-sensing algorithms, which suggests that more accurate climate models would predict even

stronger NPP declines in the future than what they currently predict.

To the best of my knowledge, the claim and the arguments that support it are novel.

The paper will be of interest to many, including in the fields of oceanography, biogeochemistry, climate-modelling, as well as policymakers (if they understand the conclusions).

I commend the authors for good work in that the manuscript is clear, short, and well-structured, and the figures and supplementary material are generally adequate and help to understand the main story.

However, I think many minor things could be improved.

Hence, in my opinion, this work is worthy of publication in *Communications Earth & Environment* after some revisions.

The most important revisions the authors should consider are the following in my opinion:

1. General presentation improvement to emphasize the science over the statistics (some details below)
2. Improve and merge Fig 4 into Fig 1 (Fig 4 is the central Figure but can be greatly improved)
3. Add a paragraph of discussion of the caveats of the method, which is missing. I am unsure what the biggest caveats are, if any, but one issue that I think needs at least a sentence is that the central claim hinges on a comparison of CMIP6 climate models with a suite of remote-sensing models that are afflicted by large uncertainties themselves. While the authors acknowledge and discuss these uncertainties, they don't discuss their effects on the conclusions drawn. In a ideal world, one would directly compare the climate models' NPP to observations of NPP and avoid the need for remote-sensing algorithms altogether. However, remote-sensing NPP estimates are the best tool we currently have for estimating NPP with global coverage from variables observed by satellites. To me, this begs the question: How would systematic bias in the suite of remote-sensing models used in the authors' analysis affect their conclusions? Another set of caveats may lie in the choice of environmental drivers, which seems arbitrary to some extent. What other drivers could have been included? What important driver could be missing, if any? Or is there reasonable confidence that the SST, CHL, and MLD set is optimal?
4. If possible, the manuscript would be greatly improved by some brief discussion on what could actually be done to improve climate models (and remote-sensing algorithms) to achieve better consensus in NPP estimates and projections. Maybe these papers could guide this discussion:
 - Henson et al. (2022; <https://www.nature.com/articles/s41561-022-00927-0>)
 - Boyd (2015; <https://www.frontiersin.org/journals/marine-science/articles/10.3389/fmars.2015.00077/full>)
5. Improvements to the Methods section (particularly the part on Multiple Linear Regressions and Earth Mover's Distance).

Below I detail all my suggestions (except for points 3. and 4. above) in order of appearance in the paper, including much more minor issues.

- L14: Some would contend that NPP is not a "major" flux when compared to other fluxes and I think the most important part here is that NPP sustains ecosystems anyway, so what about starting with it, e.g., "... (NPP) supports critical ecosystem services and is important for the carbon cycle".

- L22: I think I understand that the authors want to hint that they don't just use yearly-maximum MLDs, but everything is "seasonal" by nature in the ocean. What about removing "seasonal" and just use "mixed layer".

- L23–25: It seems obvious to me that a "model ranking scheme" is "able to sort models" and I don't think it is useful to say here in the summary that it can reduce across-model variance. What about something simpler and punchier like: "These rankings suggest that a future decline in global NPP is more likely than not and that this decline is currently underestimated by all climate models."

- L25–28: This sentence is a little unclear to me. What about splitting it into something like: "In addition, we find that models tend to statistically underestimate the NPP decline driven by sea surface temperature (SST) warming. This suggests that more accurate climate models that capture this higher SST sensitivity would predict even greater NPP declines in our warmer future climate." (I would remove the redundant "with important consequences for the marine ecosystems" since NPP was already said to support ecosystems in the first sentence of the summary paragraph.)

- L31–36: I don't think it is entirely correct to say NPP supports ecosystem services by sustaining biodiversity. In addition, I don't think that the role that NPP plays in the carbon cycle is important in this paragraph, which is about the importance of NPP for ecosystems and its uncertain future. So what about starting with that instead, with something along the lines of: "Marine NPP by phytoplankton sustains biodiversity and is essential to ocean ecosystems, but its future is uncertain." And then dive into the details of this uncertainty and the urgency of dealing with it.

- L41: I'm not sure that calling NPP a "boundary condition" is correct, but more importantly, I don't think it helps to understand this sentence anyway, so what about: "(...) utilise NPP projections from only two climate models (...)", which is a bit shorter, too?

- L46–49: This sentence is a bit long and contains redundancies, and although it has been used elsewhere, I don't think "emergent constraint" is correct or useful here (the changes and relationships are emergent, but the constraints are not, even if using some relationship as a constraint is novel). What about something like: "Remote-sensing estimates of NPP over the contemporary period (1998-2023) provide global constraints for Earth system models. In addition, emergent relationships between changes in NPP and concomitant changes in ocean environmental variables over the contemporary period provide

further constraints for Earth system models."

- L50: Remove "similarly".

- L54–57: While I try to commend the efforts of fellow researchers as often as possible, I don't think this part of the manuscript is the right place for it. It is also unclear which part has been addressed by OC-CCI. I could be wrong, but my understanding is that OC-CCI merges all the "raw" satellite data (including light but also some derived products such as chlorophyll) but not NPP. If I'm correct, then OC-CCI addresses the issue of the time period and the data being used (the first and third items in the previous sentence), in which case it would be clearer to explicitly say so in the manuscript (otherwise the reader is left wondering what OC-CCI addresses). Hence, what about: "Sensitivity to the time period or the data being used has been recently addressed by the publication of a coherent multi-sensor satellite record spanning 1998–2023 that merges all available single-sensor satellite missions with substantially reduced inter-sensor biases." (I would remove the following sentence: "The outcome is (...)".) This would also flow better logically with the following "Intrinsic differences in trends are however still expected from the range of algorithms available for quantifying NPP."

- L59: Remove "that represent a range of different approaches to derive NPP" since this is clear from the previous sentence.

- L60–63: This list of 4 algorithms confused me at first because I was expecting 6 instead. I think it would be best if the 6 algorithms were defined here, which would avoid making the reader stumble on first read of "Lee-AbPM and Silsbe-CAFE" L68, since these are not defined at this stage in the manuscript. I would also recommend avoiding the single quotes here. E.g., what about: "These algorithms include two vertically generalised production models (Eppley-VGPM and Behrenfeld-VGPM), (and so on...)"

- L64–64: I would remove the obvious "Whilst each algorithm possesses different uncertainties and caveats for estimating NPP" and start the sentence with "None of algorithms has been found (...)" ("singular" is unnecessary and may be confusing).

- L71–74: This sentence is a bit confusing and uses slightly imprecise language in my opinion. What about: "We ranked 15 CMIP6 Earth system models according to their ability to capture the emergent contemporary relationships between NPP and environmental variables (sea surface temperature, chlorophyll-a, and mixed layer depth) observed in the 6 remote-sensing algorithms." I think saying these relationships are "mechanistic" here was too much of a stretch, given these relationships are more akin to simple correlations. In addition, "parallel" is a little imprecise and the concomitance of the compared relationships can be delegated to the Methods section.

- L74–78: This sentence is a bit convoluted and would probably read better if it started with the 4 rankings that "agree" (the word "bifurcation" is probably not the best here either). What about: "Four algorithms (which includes the best performing algorithms according to XXX; Lee-AbPM and Silsbe-CAFE) concur that climate models projecting greater NPP declines rank higher, while the remaining two (Eppley-VGPM and Behrenfeld-CbPM) rank models that project slightly positive NPP trends higher." (about the "XXX" above: I would be explicit about what makes Lee-AbPM and Silsbe-CAFE better performers; I think the authors are referring to the round robin here, as they do L232, but I am not entirely sure. Please confirm)

- L78–80: This "assessment" sounds a little vague here. What about something more factual: "Furthermore, using the Lee-AbPM and Silsbe-CAFE algorithms also produce the most effective rankings (effectiveness is quantified by the reduction of inter-model variance when discarding lower ranking models)."

- L80–81: NPP decline is always likely. What about something stronger (and that repeats the same language of "decline" rather than "loss"; repetition is good here): "These results suggest that future NPP decline is more likely than not, and this decline is currently underestimated by even the best ranked CMIP6 models, which predict the most intense NPP declines."

- L90 but also L93, L98, L108, L110, L113, the "S" before the Figure number is missing in "Supplementary Figure X".

- L91 and throughout, in my opinion, there is a bit too much importance given to p-values versus the actual science or mechanism being discussed. For example, here in L91, the more important bit of information is that the increases in NPP are small. Maybe this is my personal aversion to statistical jargon, but I think that most of the p-value mentions should be relegated to Figure captions or supplementary Tables so that the main text is focused on the main message. Another issue I have is that I am not sure that I can formulate the null hypothesis that these p-values are based on in some (if not most) instances, which means that I am unable to truly interpret their meaning anyway (but, again, this could be just me).

- L101–116: The statistical part of this paragraph is a bit confusing to me. Maybe it could be streamlined a little to emphasize the science instead of the statistical tests? It would also maybe be useful to move the last sentence up to the start of the paragraph.

- L118: What about "concomitant" in place of "parallel"?

- L125–127: What about: "To statistically assess what locally drives changes in NPP, we use multiple linear regressions of contemporary trends in NPP against 4 environmental and biological drivers, for each remote-sensing algorithm."

- L127: I think it is important here to mention that warming SST can drive NPP in both directions. Increased stratification means less nutrient supply and thus NPP decline, while increases in metabolic rates are generally expected to increase NPP. One of the reasons I think this is important is because I have done a similar driver-decomposition exercise recently myself and I found that the compensation between warming (enhancing production) and the decline in nutrient supply was quite strong for my model (see Pasquier et al., 2024, Fig. 1, <https://bg.copernicus.org/articles/21/3373/2024>, but please note that I do not think the authors should cite me here)
- L132–142: What about something shorter, less detailed, and more to the point. For example, for the sentence starting L132, something like: "Using all four drivers significantly improved the multiple linear regressions for all remote-sensing algorithms." I would recommend keeping the gist of which remote-sensing algorithms had the most skillful regressions and move the statistical details (p values, R^2 values, and co) to the supplementary information.
- L144–161: I think this paragraph on coefficients needs reworking. In particular, the main results must stand out and be placed upfront. In my opinion, the most important is that NPP is driven predominantly by SST, then CHL, then MLD. The second most important (which should therefore be discussed after the main point) are the spatial distributions and the mechanistic interpretations.
- L165–167: I would not say that things "can be" done when things "have been" done. What about something like: "We apply the same multiple linear regression of NPP against SST, CHL, and MLDs to 15 CMIP6 Earth system models and rank these models according to their capacity to capture the emergent relationships observed with the remote-sensing algorithms and data. Specifically, we (...)"
- L175–189: As for the similar paragraph on remote-sensing regressions, I would start with the most important point, which is that the coefficients are different in magnitude globally, and then move to the more detailed discussion of the distributions. The authors should also consider discussing the mechanistic relationships that are explicitly built in these models, in the same way that Fig. S1 shows the built-in relationships of NPP with input variables for remote-sensing algorithms. In biogeochemistry models, NPP is explicitly related to temperature and chlorophyll as far as I know, and my intuition is that these relationships would heavily influence the regressions. I guess this might also help some interpretations.
- L194: I would recommend hand-holding here to explain what high/low EMD means, maybe simply a parenthesis with something like: "(low EMD means good agreement and thus high rank)" (but maybe it is the other way around, or maybe worse I misunderstood completely).
- L205: Add "Earth system" in "between remote sensing and Earth system models for these two variables"
- L214: As much as I like short, clear, strong statements, I think this one is a bit too strong, and I think it is best to say which way Z scores improve ranking rather than the other way around. What about: "A low Z score thus indicates that the NPP–driver relationship in the Earth system model matches that of the remote-sensing algorithm well".
- L215: Does "combine" here mean "sum"? If yes, I would suggest using "sum" and remove "using equal weighting".
- L218: algorithms don't "manage" to reduce Δ NPP standard deviation. It would also help to reiterate what reducing across-model variance implies here. What about: "Only for the Eppley-VGPM, Behrenfeld-VGPM, Lee-AbPM, and Silsbe-CAFE algorithms does removing low-ranking Earth system models significantly reduce the across-model variance of Δ NPP, indicating more effective ranking (ref)." (I would then remove the sentence L225–227)
- L227–231: I would rephrase this as something simpler like: "The Lee-AbPM and Silsbe-CAFE algorithms both produce the most effective rankings and rank Earth system models with negative future NPP predictions the highest." and remove "The remaining algorithms do not display any marked divergence in Δ EMD mean or standard deviation"
- L235: Given the suggestion above that contains part of this sentence, I would rewrite as: "Together this suggests a greater likelihood of global NPP decline in the future."
- L268: What about: "Remote sensing is a powerful tool".
- L273: There is one reference but this sentence mentions previous studies (plural). Maybe the authors meant to add more references here?
- L279: What is "the resource limitation diagnostics in Earth system models"? Is there a reference for it?
- L304: Is the code available publicly? (E.g., on a public repository such as GitHub, or better yet, in a public archive such as Zenodo.)
- L362+ Methods section on MLR and EMD: I find this section quite hard to read with a number of occurrences of imprecise or convoluted wording. I think more equations and symbols here would help navigate the rather complicated assemblage of metrics. For example, among other things, I wonder if "normalized to the mean along the time dimension" means "normalized by the time-mean". I also wonder what checks and tests were conducted. I wonder what a significant pixel is. Equation 1 is not displayed correctly (I see a dotted square in the integral). "l1" on the left-hand-side of Equation 1 is not

defined. Equation 1 also looks like it is missing a sentence to introduce it. The sentence just after Equation 1 starts with "Where" with an upper case "W" when it should be a lower case "w", right? By "proportion" I think the authors mean "area" but I am not entirely sure. The "A" in "A mean and standard deviation was calculated" is strange, as "mean" and "standard deviation" are well-defined. Statements like "x is the value of the model" is obscure (what model? the value of what?). I don't mean to be disparaging with the series of critiques above but I do think that the authors should clarify this section so that any interested reader can understand the details of the methods employed and reproduce each step.

- Fig 1:

- While I understand that the authors computed "decadal trends of annual means", this sounds equivalent to simply "mean decadal trends".

- What is the normalization used for NPP trends?

- Fig 4: This figure is central to the manuscript, yet I think it could be improved a fair amount. I understand the intent of the authors to visualize the Δ NPP along the rankings, but these bar plots are all redundant with Fig 1. In addition, I simply find this Figure painful to grasp at a glance, as it forces the reader to keep looking back and forth at the legend and to squint to distinguish colors. Furthermore, I think that the rankings themselves are a little misleading, in the sense that it does not show the Z score. As a solution to these issues, I would consider merging Fig 1 and 4 in the following way: First, sort the Earth system models by Δ NPP instead of alphabetically in Fig 1. (This is to prepare the merge with Fig 4 but it will also help with spotting the disagreements between NPP trends and Δ NPP.) Then, append a 3rd panel (panel c) at the bottom containing a heatmap (see, e.g., https://matplotlib.org/stable/gallery/images_contours_and_fields/image_annotated_heatmap.html) of the Z-scores (align the columns with the Earth system models of panels a and b, and use the rows for remote-sensing algorithms, also sorted by NPP trend.) By choosing a colormap for the heatmap that highlights the models that rank best, this will show at a glance the central message of the paper, add extra useful information visually (the Z scores), all while removing 1 Figure with 6 redundant panels. It will also place the central message in the first Figure, which is nice on the readers that get tired quickly. If the authors do follow this suggestion, they should make sure that the sorting of Earth system models is applied to all Figures to avoid confusion.

- Fig S1: y-axis label mentions "normalized NPP". What this normalization is should be explained in the caption.

- All the other figures are beautiful.

Communications Earth & Environment is committed to improving transparency in authorship. As part of our efforts in this direction, we are now requesting that all authors identified as 'corresponding author' create and link their Open Researcher and Contributor Identifier (ORCID) with their account on the Manuscript Tracking System prior to acceptance. ORCID helps the scientific community achieve unambiguous attribution of all scholarly contributions. You can create and link your ORCID from the home page of the Manuscript Tracking System by clicking on 'Modify my Springer Nature account' and following the instructions in the link below. Please also inform all co-authors that they can add their ORCIDs to their accounts and that they must do so prior to acceptance.

Version 1:

Decision Letter:

Dear Dr Ryan-Keogh,

Your manuscript titled "Global decline in net primary production underestimated by climate models" has now been seen by our reviewers, whose comments appear below. In light of their advice we are delighted to say that we are happy, in principle, to publish a suitably revised version in Communications Earth & Environment.

We therefore invite you to revise your paper one last time to edit your manuscript to comply with our format requirements and to maximise the accessibility and therefore the impact of your work.

EDITORIAL REQUESTS:

*****Please take care to match our formatting and policy requirements. We will check revised manuscript and return manuscripts that do not comply. Such requests will lead to delays. *****

SUBMISSION INFORMATION:

OPEN ACCESS:

Communications Earth & Environment is a fully open access journal. Articles are made freely accessible on publication. For further information about article processing charges, open access funding, and advice and support from Nature Research, please visit <https://www.nature.com/commsenv/open-access>

Link Redacted

Best regards,

Alice Drinkwater, PhD
Associate Editor
Communications Earth & Environment
@CommsEarth

REVIEWERS' COMMENTS:

Reviewer #1 (Remarks to the Author):

Dear editor and authors,
the authors have thoroughly addressed all the concerns raised in the first review round, and their revisions have strengthened the paper significantly.
I recommend the paper to be accepted as it is.
Best regards,
Etienne Pauthenet

Reviewer #2 (Remarks to the Author):

The reviewed manuscript addressed all the questions, especially my main concerns on the satellite NPP comparison the methods clarification. The authors also justified the few modifications that were not included in the manuscript. After going

through the manuscript once again, I don't have any more comments or suggestions to the authors.

Response to Reviewers - Global decline in net primary production underestimated by climate models

We would like to thank all the reviewers for their time and effort in providing this invaluable feedback on our manuscript. We have provided responses to each comment as text tabbed to the right denoted in Times New Roman as **RX.X** dependent upon reviewer number and reviewer comment. Due to the broad impacts of certain responses to specific reviewer comments and the overlapping nature of several comments e.g. the impact of methodological approach, the interpretation of results and format, we first summarise here the major changes that emerged before moving on to addressing the individual points in more detail.

1. All of the remote sensing analyses have been revised and are now performed using Jackknife resampling of 80% of the 26 time series, where we have 7 simulations each of:
 - a. Annual mean trends in remote sensing NPP
 - b. Area-weighted mean-normalised trends in remote sensing NPP
 - c. MLR coefficients of drivers of remote sensing NPP
 - d. EMD values of MLR coefficients
 - e. Earth System model ranking assessments
2. The MLR statistics now account for autocorrelation and heteroskedasticity and report the adjusted R^2 value.
 - a. This change in methodology revealed the multicollinearity between MLD_{max} and MLD_{min} so instead, we switched to using only the annual mean MLD - more simply referred to as MLD throughout the manuscript.
3. The outlier detection method was switched from excluding the bottom 10th and top 90th percentile data to instead using a IQR fence test, with the IQR factor set at 3 (double the normal factor usually applied for detecting outliers).
4. Discussions pertaining to the uncertainties of the remote sensing algorithms have now been moved to a new dedicated section with the heading “Assessing the merits of remote sensing algorithms”, which includes :
 - a. The primary determinants of NPP for each algorithm (Original Figure S2).
 - b. The round robin exercise results which have been expanded to include additional studies (Original Table S1).
 - c. The jackknife trend analysis which has been expanded to provide coefficients of variation of the trends (Original Figure S3).

Reviewer #1 (Remarks to the Author):

This is an important paper comparing marine Net Primary Production (NPP) from Earth System Models (ESM) and from satellite observations over the contemporary period (1998-2023). First the trends of NPP are presented, revealing that the observed NPP is declining and that the ESM are either not predicting or are underestimating this decrease. Six different remote sensing algorithms are used and compared, and fifteen ESMs are presented.

To evaluate which ESMs is modelling the best response to environmental drivers of NPP, multi linear regressions are computed between NPP and the drivers of NPP which are SST, chlorophyll-a and annual mean and max of MLD. This MLR is also done on observations for the six different algorithms and observed SST, chl-a and MLD. The resulting coefficients of regression of the fifteen ESMs are compared to the ones of the six remote sensing algorithms with an Earth movers' distance (EMD) metric, from which they extract a Z-score. A high Z-score indicates a poor relationship between NPP drivers.

The authors state that the ranking of the ESMs according to this Z-score confirms "a strong likelihood of negative future projections in NPP". The figure 4 and 5 are however not showing such a clear message. I am not convinced that there is a "strong" likelihood of negative future projections in ESMs' NPP according to this analysis. This paper would require to either tone down the message, or to bring "strong" elements to confirm this conclusion.

In more details :

1) The main result is that the "global decline in NPP is underestimated by climate models" over the contemporary period. This result is shown with the barplots of trends in % decade⁻¹, in figure 1 for the ESM and figure S2b for the remote sensing algorithms. I am not convinced that the model ranking scheme is adding valuable insights to support this result, contrary to what is stated in the abstract L24 : "This scheme is able to sort models and in so doing reduce across-model variance with results suggesting that future declines in global NPP are more likely and currently underestimated." "reduce across-model variance" is not that clear on the figure 4 and 5 :

R1.1 We no longer include the figure that refers to cross-model variance. Looking at global trends in NPP averaged across the model ensemble (which is the norm) generates a mean trend that is close to zero (because some are positive and some are negative). This is indicative of very little change in global NPP. With the above mentioned adjustments to the methodological approach 5 out of the 6 algorithms now rank models with negative

future trends highest. Even the largest negative trends in NPP from any of the earth system models to 2100 (-0.2%) is substantially smaller (nearly one order of magnitude) than the more robust satellite NPP algorithm trends over the contemporary period (-1.5%). Combined these three points suggest that 1) ensemble means of no change are unlikely to be true, 2) future declines in global NPP are more likely and 3) those models that predict global declines in NPP are nonetheless likely to be underestimated.

- In figure 4, there is only three out of six remote sensing algorithm that are ranking the strong decline in NPP first, the three others are a bit random, Eppley-VGPM and Behrenfeld-CbPM are even ranking the strong declines tenth or higher.

R1.2 With the above mentioned adjustments to the methodological approach 5 out of the 6 algorithms now rank models with negative future trends highest. See revised Figure 4.

- In figure 5a, if we retain eight models out of fifteen for example, the DeltaNPP 1sigma increased for four of the remote sensing algorithm, but it should be decreasing if the ranking was selecting the models that are predicting a decrease of NPP.

R1.3 We no longer include this figure in the manuscript due to the change in our methodological approach. However, we would like to point out that the $\Delta\text{NPP } 1\sigma$ did indeed decrease for the ranking of models that predicted a decrease in NPP. We concede that perhaps the orientation of our axis going from 15 to 1 was misleading.

- In figure 5, if we keep only the four best ranked models, two algorithms show no decrease in DeltaNPP 1sigma and three algorithms show a positive DeltaNPP mean or equal to zero.

R1.4 Please note that the change in our approach for the ranking exercise means that this is no longer the case. Instead what we find is that our jackknife resampling ranking exercise now concludes that 5 out of the 6 algorithms all rank negative projections higher, and only Eppley-VGPM ranks positive projections higher. See below for the revised figure.

“Figure 4: Ranking Earth system models using Z-score assessments of the Earth mover’s distance metric. Bar plots of mean±standard deviation Jackknife resampled ranked Earth system model ΔNPP (Pg C year⁻¹) for (a) Eppley-VGPM, (b) Behrenfeld-VGPM, (c) Behrenfeld-CbPM, (d) Westberry-CbPM, (e) Lee-AbPM and (f) Silsbe-CAFE NPP algorithms. All bars are coloured by the mean Z-score across the jackknife resampling exercise. Please note that the absence of an errorbar is indicative of the same model being ranked in the same position for all 7 of the jackknife simulations.”

- Finally I do not see how the model ranking scheme is suggesting that the global NPP decline is underestimated, as stated in the abstract in the same sentence. Could the authors please explain further this statement?

R1.5 We apologise if this point was not made as clear in the manuscript and we will try to elaborate here. Future trends are underestimated because the model ensemble is averaging across models that predict both positive and negative trends and thus the net future global trend averaged across all models is close to zero. Our approach attempts to rank the models based on their ability to reflect driver response relationships observed

between satellite NPP and CHL, SST and MLD. This approach uses 6 different algorithms to rank the models based on the different satellite NPP data products being used. With 5 of the 6 algorithms ranking models that predict future declines in NPP highest. The study moves on to then evaluate the merits of the different satellite NPP algorithms and determines that the Lee-AbPM and Silsbe-CAFE are the best due to:

1. Their performance in in situ NPP round robin exercises (generally lower RMSD compared to other algorithms)
2. The insensitivity of their trends using jackknife resampling (no uncertainties in direction of trend, just small uncertainties in the magnitude of trends)
3. Their alternative approach to estimating NPP that moves away from using absolute quantities of photosynthetic pigments and instead uses the efficiency of light absorption.

Despite an agreement in the direction of future NPP trends from the top performing satellite NPP algorithms and the highest ranked earth system models, the Lee-AbPM and Silsbe-CAFE trends over the contemporary period are much larger (up to an order of magnitude) than any of the projected trends to 2100 from any of the CMIP6 Earth system models. The MLR analysis indicates that one of the primary reasons for this is most likely ocean warming, as evidenced by the negative coefficients (i.e., increasing SST results in declining NPP). The same MLR analysis when performed on the CMIP6 Earth system models shows that SST coefficients that are either very close to 0 or only mildly negative, meaning that the models are not correctly parameterising the potential negative impacts of warming on phytoplankton NPP. This is perhaps not surprising given the way in which many models parameterise phytoplankton growth, i.e. often relying on an Eppley growth curve function which dictates an increase in NPP with increasing temperature. We argue that if we alter this parameterisation to allow models to be more sensitive to the negative impacts of warming on phytoplankton growth then NPP projections (particularly those from the top ranked ESM's) will begin to show larger declines that align in magnitude with those of the top performing Satellite NPP trends. This parameterisation adjustment will not only impact 'business as usual' scenarios with large projected increases in ocean temperature, but also high mitigation scenarios which still have some warming locked in due to the feedback mechanisms of emitted CO₂. As such we argue that the projected no change in NPP under the high mitigation scenarios is underestimated because they are not accounting for the negative impacts of increasing ocean temperatures.

2) How is it possible to compare the contemporary period of the ESMs and of the remote sensing data? I believe the ESMs are not forced with real data so the ESM years do not coincide with observed years?

Also, over which period is the MLR computed for the ESMs? Is it over 1998-2023 or over 1850-2100? I did not find this information in the paper, could the authors please explain further?

R1.6 We are not directly comparing the period of the contemporary trends of ESMs with the period of trends observed from remote sensing data, we simply highlight how the “observed” remote sensing trends are much larger than the Earth system model projected trends (despite the trajectory of ESM trends being to the end of the century). Following prior work and conventions, we performed the MLR analyses on the ESMs trends over the longer term historical period (i.e. from 1850-2014) to be consistent with prior modelling exercises. The reasoning for this is that the reliability models can only be assessed by observations of the past and present. This means that models are assessed against criteria that are not necessarily informative in terms of the quality of model projections of future climate change. The emergent constraint approach attempts to address this problem by identifying robust, physically interpretable relationships between Earth system feedback behaviours on well-observed timescales (Eyring et al., 2019, Nature). The fidelity of an emergent constraint, thus, depends on the correlation of the relationship and the uncertainty of the observations (Terharr et al., 2023, Science Advances).

Our approach attempts to determine how close the relationships parameterised in ESMs, e.g. the impact of temperature changes on phytoplankton productivity, reflect those observed from remote sensing (e.g. the relationship between warming and NPP). Of course, the remote sensing dataset is only available over the contemporary period, but that does not mean that the inherent relationships between changing variables (e.g. SST, CHL, MLD and NPP) cannot be interrogated to see how well those relationships are reflected in ESMs.

We have amended the sentence on line 145 (originally line 168) to include the years of the historical period, 1850-2014. Please also note that following revisions of the methods for the MLR, we have now switched from using MLD_{max} and MLD_{min} to instead using annual mean MLD. This was due to issues of multicollinearity between MLD_{max} and MLD_{min} .

Line 158: “MLR analysis between modelled NPP and the three associated drivers (SST, CHL and MLD) were repeated globally for each Earth system model over the historical period (1850-2014) but without jackknife resampling.”

3) In figure S8 we see large variations in the Z-score. It would be interesting to display these variations against DeltaNPP, to add more information than the ranking. If the

strongly decreasing ESMs have way smaller Z-scores than the other it would be a strong point for the paper. Could the authors display the Z-score in a scatterplot against DeltaNPP? Like a combination of Figure 4 and Figure S8. That would be 6 scatterplots on which we should discriminate the better scoring ESMs.

R1.7 We thank the reviewer for this suggestion and we have plotted the requested figure below. However, on reflection, we do not feel that these types of plots are any easier to interpret than our original approach. The Z-scores are relative to each remote sensing algorithm, where each algorithm places a different relative weighting of the proposed drivers.

Please note however that we have now amended Figure 4 to bring in the Z-score information, and we have deleted Figure S8. Below is the new Figure 4, where we have also amended the methodological approach for how we performed the MLR and EMD analysis. Due to the high degree of variability in trends evident in the Jackknife analysis

we modified our statistical approach to capture this variability. The MLR analysis is now performed on Jackknife simulations, i.e., 80% of the 26 year time series representing 7 simulations, to generate sets of coefficients for each simulation. Each simulation's coefficients in turn are then compared to the ESM coefficients to generate a range of EMD values and subsequent rankings. The new Figure 4 rankings below represent the mean \pm stdev of each of the 7 ranking exercises, coloured with the mean Z-score from the 7 ranking exercises.

We hope that this new figure provides the level of information you had originally requested, so that the readers do not need to refer to multiple figures as previously. Whilst also providing a clearer picture of which algorithms perform best for the ranking exercise.

“Figure 4: Ranking Earth system models using Z-score assessments of the Earth mover’s distance metric. Bar plots of mean \pm standard deviation Jackknife resampled ranked Earth system model Δ NPP (Pg C year⁻¹) for (a) Eppley-VGPM, (b) Behrenfeld-VGPM, (c) Behrenfeld-CbPM, (d) Westberry-CbPM, (e) Lee-AbPM and (f) Silsbe-CAFE NPP algorithms. All bars are coloured by the mean Z-score across the jackknife

resampling exercise. Please note that the absence of an errorbar is indicative of the same model being ranked in the same position for all 7 of the jackknife simulations.”

In line comments :

L245-246: "modelled trends of NPP over the contemporary period remain too low relative to remote sensing estimates" this statement is supported by figure 1b and figure S2b. It is the main result of this paper (title), could the authors add in figure 1 a column for observed NPP trend and DeltaNPP, next to the "ALL" column? To display 6 barplots (for the 6 algorithms) of trends and differences over the contemporary period that are comparable with the ESMs. Because for now we have to read supplementary figure S2b to assess the main result.

R1.8 Unfortunately it is not possible to calculate a Δ NPP from remote sensing data, because the length of our time series is only 26 years, compared to ESMs which are looking at the difference over ~80 years. Additionally we have opted not to include the contemporary trends from remote sensing in this figure as it detracts from the main point we are trying to make here, that we have a wide range of projections and it is difficult to assess which is the most likely scenario. We would like to point out though that we have now changed the figure to be ordered by Δ NPP values, please see figure below. If we include the % NPP trend from satellites onto this figure as is suggested then we cannot see the intermodal variability as the y axis range would have to change by an order of magnitude from +/- 0.2 to +/- 2. However, in order to be able to better illustrate this point (i.e. that the trends in contemporary NPP from algorithms are typically much larger than those predicted by earth system models over either the contemporary or future) we refer to Figure S1 which has the mean normalised NPP trend per biome and global for each satellite algorithm (with a y axis of $\pm 2\%$ year⁻¹)

“Figure 1: Variability of net primary production trends from CMIP6 Earth system models. (a) Area-weighted mean-normalised net primary production (NPP) annual mean trends (% year⁻¹) calculated using ordinary least squares for the historical (1850-2014), contemporary (1998-2023) and future (2015-2100) periods for the CMIP6 Earth system model ensemble. (b) Area-weighted Δ NPP (Pg C year⁻¹), calculated as the difference between the end of the historical period (1995-2014) and the end of the century (2081-2100), for each of the Earth system models in the CMIP6 ensemble. Both panels are sorted by Δ NPP from low to high values.”

Figure S1: Comparing trends of net primary production from different remote sensing algorithms across biomes. (a) map of ocean biomes defined by Fay & McKinley²⁷ as the North Pacific (NP), Equatorial Pacific (PEQU), South Pacific (SP), North Atlantic (NA), Equatorial Atlantic (AEQU), South Atlantic (SA), Indian (IND) and Southern Ocean (SO) including the ice (ICE), subpolar seasonally stratified (SPSS), subtropical seasonally stratified (STSS) and subtropical permanently stratified (STPS) regions. White pixels are regions which could not be classified into a biome. (b) Bar plot of jackknife resampled area-weighted mean-normalised annual mean trends in net primary production (NPP; % year⁻¹) calculated using ordinary least squares per ocean biome for the Eppley-VGPM, Behrenfeld-VGPM, Behrenfeld-CbPM, Westberry-CbPM, Lee-AbPM and Silsbe-CAFE algorithms.

L70: Could the authors introduce briefly what differentiate Eppley, Behrenfeld or Westberry algorithm? There is only a short description of the VGPM and CbPM (L61-62) but why is there 4 of them with different prefixes?

R1.9 We thank the reviewer for this suggestion and we have added the following text to the main document to better elaborate on the algorithm differences:

Line 56: “Here we focus on six algorithms including: (1) the ‘vertically generalised production model’s (Eppley-VGPM¹⁶ and Behrenfeld-VGPM¹⁷), which define phytoplankton growth as a function of chlorophyll-a, light and temperature, the difference being that Eppley-VGPM is an exponential function of temperature, while Behrenfeld-VGPM is a 4th order polynomial; (2) the ‘carbon-based production models (Behrenfeld-CbPM¹⁸ and Westberry-CbPM¹⁹), which incorporate particulate backscatter as a proxy for phytoplankton carbon but differ in that Westberry-CbPM is both depth and wavelength resolved whilst Behrenfeld-CbPM is not; (3) the ‘absorption-based production model’ (Lee-AbPM²⁰), which defines NPP as a function of phytoplankton absorption rather than chlorophyll; and (4) the ‘carbon, absorption, and fluorescence euphotic’ resolving model (Silsbe-CAFE²¹), which integrates the learning from all the above algorithms to define NPP as a function of energy absorption and efficiency (for more details please see Methods).”

We have also expanded the methods to provide the general equations for each algorithm:

Line 324: “Net primary production (NPP; mg C m⁻² d⁻¹) was calculated using the following algorithms, the ‘vertically generalised production model’s (Eppley-VGPM¹⁶ & Behrenfeld-VGPM¹⁷; Equation 1), which relies on the relationship between chlorophyll *a* and temperature derived growth rates; the ‘carbon-based production model’s (Behrenfeld-CbPM¹⁸; Equation 2 & Westberry-CbPM¹⁹; Equation 3), which uses backscatter derived phytoplankton carbon as a biomass indicator and physiology derived as variability in the chlorophyll *a* to carbon ratio; the ‘absorption-based production model’ (Lee-AbPM²⁰; Equation 4), which does not make any assumptions on either chlorophyll *a* or backscatter as biomass proxies but relies on absorption characteristics to infer phytoplankton photosynthetic efficiency; and the ‘carbon, absorption, and fluorescence euphotic’ resolving model (Silsbe-CAFE²¹; Equation 5), which derives NPP as a function of energy absorption and efficiency.

Equation 1:
$$NPP = Chl \times f(T) \times DL \times f(PAR) \times Z_{eu}$$

Where Chl is chlorophyll-a concentration, $f(T)$ is a temperature based growth function (exponential for Eppley-VGPM and a 4th-order polynomial for Behrenfeld-VGPM), DL is day length in hours, $f(PAR)$ is the parameterized light term for the ratio of realised NPP to maximum potential NPP and Z_{eu} is the depth of the euphotic zone.

Equation 2:
$$NPP = C_{ph} \times \mu \times I_g$$

Where C_{ph} is phytoplankton carbon derived from an empirical relationship with particulate backscatter at 443 nm ($b_{bp}(\lambda_{443})$), μ (μ_{max}) is the growth rate and I_g is the growth irradiance term.

Equation 3:
$$NPP = \int_0^{Z_{NO_3}} \mu \times C_{ph}(z) \times f(I_g) \times dz$$

Where Z_{NO_3} is the depth of nitracline, defined as the depth at which nitrate + nitrite exceed 0.5 μ M, and dz is the depth.

Equation 4:
$$NPP = a_{ph}(\lambda_{443}) \times K_d(\lambda_{490}) \times PAR$$

Where $a_{ph}(\lambda_{443})$ is phytoplankton specific absorption at 443 nm, $K_d(\lambda_{490})$ is the light attenuation coefficient at 490 nm and PAR is the daily available photosynthetic radiation.

Equation 5:
$$NPP = \epsilon \times \mu \times h(\epsilon / (a_{ph}(\lambda_{443}) + K_d(\lambda_{490})))$$

Where Q_{PAR} is energy absorption, $\phi_{\mu}^{\square\square}$ is efficiency at which the absorbed energy is converted into carbon biomass, E_k is the light saturation parameter and E is daily PAR. For more details on all equations please refer to their specific publications.”

Figure 4 :

There is IPSL-CM6A-LR twice in the legend instead of MPI-ESM1-2-LR, could the authors double check that the colors correspond to the ESMs name?

R1.10 We would like to thank the reviewer for finding this error. However following the changes we have made to the analysis this issue no longer persists. The revised Figure 4 is presented below where each bar is instead now coloured with the mean Z-score from the Jackknife resampling ranking assessment.

“Figure 4: Ranking Earth system models using Z-score assessments of the Earth mover’s distance metric. Bar plots of mean±standard deviation Jackknife resampled ranked Earth system model ΔNPP (Pg C year⁻¹) for (a) Eppley-VGPM, (b) Behrenfeld-VGPM, (c) Behrenfeld-CbPM, (d) Westberry-CbPM, (e) Lee-AbPM

and (f) Silsbe-CAFE NPP algorithms. All bars are coloured by the mean Z -score across the jackknife resampling exercise. Please note that the absence of an errorbar is indicative of the same model being ranked in the same position for all 7 of the jackknife simulations.”

Reviewer #2 (Remarks to the Author):

Summary

The manuscript first introduces that climate models from the CMIP6 often disagree with the future trends of NPP (including increase or decrease) in business-as-usual scenario (SSP5-8.5) as well as the current NPP methods using satellites observations. The study claims that the two satellite methods (Lee-ABPM and Silsbe-CAFE) that show the highest NPP decline (Lee-ABPM and Silsbe-CAFE) are more accurate, and their response coefficients to SST, Chl-a, and MLD, can be compared to the same coefficients in climate models to rank the climate models that possess similar responses to the contemporary period from the satellites observations. This shifts to two main arguments that the best ranked models compared to Lee-ABPM and Silsbe-CAFE are the ones showing the highest NPP declines but that even the climate model NPP highest decline is underestimated when compared to the satellite observations. The manuscript suggests that its work can be used for improving confidence in climate models.

The conclusions are novel and interesting, and the manuscript is well written in most parts. However, the evidence should be strengthened, methodological choices better justified, and methods explained in more detail. In my opinion, the work is relevant to the field and is worthy of publication but only if the major and minor comments are addressed.

Major comments

The main claim of the study (Global decline in net primary production underestimated by climate models) strongly relies on demonstrating that Lee-ABPM and Silsbe-CAFE are the most accurate methods for estimating NPP by satellite observations because only those algorithms are showing the substantial decline not captured in the climate models. However, this demonstration evidence I found in the manuscript seems rather weak. One evidence is summarised in Table S1 and it seems to include data only on tropical regions. Given the different biomes considered in the global estimation, it is necessary to show better performance in more different regions. I also think it is necessary to give a plausible explanation on the reason both Lee-ABPM and Silsbe-CAFE performs better. Strengthening this part would better support the main claim of this study.

R2.1 We thank the reviewer for raising these comments and we hope that the additional evidence we present is enough to strengthen the main claim of the approach and results. Firstly we would like to note that we do not argue that the Lee-ABPM and Silsbe-CAFE algorithms are the most accurate methods for estimating NPP simply because only those algorithms show substantial declines not captured in the climate models. Overall, we

discuss three lines of evidence around the ‘accuracy’ of different algorithms: 1) skill in intercomparison exercises, 2) robustness of NPP trends to the sampling of the time series (i.e. through 7 jackknife simulations) and 3) model complexity and the underlying drivers of the productivity calculation. This results in stronger support for the Lee-AbPM and Silsbe-CAFE algorithms. We have added further analysis on these points in the revised paper - please see below.

Intercomparison exercises: We have expanded the original Table S1 to now include evidence from additional regions and studies. In addition, we have changed the table to a figure as shown below (now Figure S8). The figure now not only shows results from BATS and HOTS, but many other regions across the globe including the Southern Ocean, North Atlantic and various coastal regions for four separate intercomparison exercises. There is still an overall tendency for Lee-AbPM and Silsbe-CAFE to report the lowest RMSD values in the majority cases (see the number of bold values and black outlined boxes for Lee-AbPM and Silsbe-CAFE relative to the other algorithms). Focussing on the most recent exercises, the only regions where Lee-AbPM and Silsbe-CAFE do NOT perform best are the coastal north east Atlantic, west Antarctic Peninsula, Arabian Sea and the pelagic North Atlantic.

Figure S8: Comparing differences between remote sensing NPP algorithms and direct field measurements. Root mean square differences (RMSD) between NPP estimated from algorithms (Eppley, 1972; Behrenfeld & Falkowski, 1997; Behrenfeld et al., 2005; Westberry et al., 2008; Lee et al., 2011; Silsbe et al., 2016) and direct field measurements from the Bermuda Atlantic Time Series (BATS), Hawaii Oceanic Time Series (HOTS), Western Antarctic Peninsula (WAP), Ross Sea, Mediterranean Sea, coastal north east Atlantic (NEA), Black Sea, Arabian Sea, pelagic North Atlantic (NABE), Antarctic Polar Frontal Zone (APFZ), California coast (CALCOFI), Mediterranean Sea (DYFAMED), Scotia Sea (AMLR), Cariaco basin (CARIACO) (Saba et al., 2010; Silsbe et al., 2016; Tao et al., 2017; Wu et al., 2024). Cells with bold text and a black border represent the algorithm which had the lowest RMSD for the specific study. Please note that empty cells means that the algorithm was not implemented during the study.

We do note that this may not cover every biome in our analysis but we currently lack a comprehensive database of in situ carbon uptake and bio-optical measurements. We have added some sentences to this effect.

Line 312: “Future assessments should not only consider the uncertainties inherent to remote sensing algorithms (Supplementary Information Fig. S8), despite the complexity in deriving them (Song et al., 2024), but should also expand on the Round Robin intercomparison exercises (Supplementary Information Fig. S8) as more in situ data becomes available.”

Robustness of trends: We refer to our jackknife resampling analysis which also points to Lee-AbPM and Silsbe-CAFE being the better performing algorithms (now Figure S9). As 1998 was a strong El Nino year we were concerned about what influence this could have on our contemporary trends, and so rather than exclude this year and shortening our time series length we used jackknife resampling to determine how resilient the trends are to time series length and start/end date. These results discussed in the submitted manuscript showed that the directional dominance of the two VGPM trends were highly susceptible to this analysis, switching from overall positive to overall negative trends globally, which thus lowers our confidence in the robustness of their trend detection. Whilst the CbPM trends did maintain their directional dominance towards negative trends, the proportion of positive to negative trends increased. Only the Lee-AbPM and Silsbe-CAFE were relatively insensitive to this resampling approach. Furthermore, we have now added additional plots to show the coefficient of variation of the trends, where $CoV = \text{Trend } 1\sigma / \text{mean Trend}$. For the VGPM and CbPM algorithms we have a high CoV globally, indicative of low agreement between the trends of the different simulations. The Lee-AbPM and Silsbe-CAFE only have high CoVs in some high latitude regions (predominantly in the Southern Hemisphere), however given the PDF distribution of the trends, which retain a dominance in negative trends) we know that this low agreement is not due to a change in direction, but rather due to differences in the absolute magnitude of the negative trends in both the Lee-AbPM and Silsbe-CAFE algorithms. Thus to conclude, the results of the jackknife simulations generate more confidence in the Lee-AbPM and Silsbe-CAFE NPP algorithms, relative to the other four algorithms.

“Figure S10: Exploring the sensitivity of trends in annual mean net primary production. Maps of coefficient of variation (a,c,e,f,g,i,l,k) and normalised probability density function (PDF) plots (b,d,f,h,j,l) of trends in annual mean net primary production (NPP; $\text{mg C m}^{-2} \text{d}^{-1} \text{year}^{-1}$) from 1998-2023 (Original) and the results from a Monte Carlo Jackknife experiment in which 20 years of the 1998-2023 period are sub-sampled for trend calculations (Jackknife Simulations) for the (a,b) Eppley-VGPM, (c,d) Behrenfeld-VGPM, (e,f) Behrenfeld-CbPM, (g,h) Westberry-CbPM, (i,j) Lee-AbPM and (k,l) Silsbe-CAFE NPP algorithms.

Coefficient of variation calculated as the Jackknife trend 1σ over the absolute mean Jackknife trends. Shaded regions in the PDF plots represent the Jackknife Simulation mean \pm standard deviation. Only pixels where the trend is significant ($p<0.05$) are included in the PDF distributions.”

Underlying drivers: Here we reflect on algorithm complexity and how well the algorithms estimate NPP and their primary drivers of variability. The original figure S1 (now Figure S7) demonstrated how both VGPM algorithms were driven by SST and chlorophyll, with PAR playing a secondary role. The CbPM algorithms are supposed to be ‘carbon-based production models’, as in phytoplankton carbon is the primary determinant of productivity. However, consistent with their studies (Song et al., 2024; Remote Sensing of Environment, doi:10.1016/j.rse.2024.114304) we find that they are in fact driven predominantly by chlorophyll over all other input variables. In essence we can then begin to state that both the VGPM and CbPM algorithms are all chlorophyll-based NPP models, i.e. productivity is linked to the quantity of phytoplankton pigments. Whereas Lee-AbPM and Silsbe-CAFE are primarily determined by phytoplankton absorption, i.e., they focus on the efficiency of light absorption for driving phytoplankton productivity. Something that has been noted in the past by Marra et al. (2007; Deep-Sea Research I) “*environmental variability is expressed through the absorption properties of phytoplankton pigments rather than their quantity*”. Yet we would note here that whilst our approaches for modelling NPP from remote sensing have evolved to this point, most Earth system models still parameterise growth using a Eppley curve (the same as is implemented in the Eppley-VGPM algorithm).

“Figure S8: Exploring the input variable dependency in estimating net primary production. Line plots of max-normalised net primary production (NPP) calculated using the (a) Eppley-VGPM, (b) Behrenfeld-VGPM, (c) Behrenfeld-CbPM, (d) Westberry-CbPM, (e) Lee-AbPM and (f) Silsbe-CAFE NPP algorithms. Input variables include sea surface temperature (SST), chlorophyll-a (CHL), photosynthetically active radiation (PAR), particulate backscattering (b_{bp}), mixed layer depth (MLD), diffuse attenuation coefficient (K_d), phytoplankton absorption (a_{ph}) and detrital absorption (a_{dg}). The input variable being tested was allowed to range between the climatological (1998-2023) 20th and 80th percentile, whilst the other input variables were held constant at the climatological median value.”

All of the above information which was previously in different locations throughout the manuscript has now been moved to 1 single section with the heading “Assessing the merits of the different remote sensing algorithms”. With this new revised text outlined below.

Line 207: “The ranking scheme developed here has resulted in a bifurcation between five algorithms which rank models that have negative NPP projections higher versus one (Eppley-VGPM) which prioritises models with positive NPP projections. Whilst both Eppley-VGPM and Behrenfeld-VGPM are primarily driven by SST and CHL (Supplementary Information Fig. S7a,b), only Behrenfeld-VGPM implements a penalty on growth when temperatures increase beyond a certain threshold, which could explain why it ranks negative projections higher, despite similarities in the regional distribution of positive trends in satellite NPP over the

contemporary period. On the other hand, Eppley-VGPM utilises an exponential growth curve with temperature to parameterise phytoplankton growth, similar to how some Earth system models represent NPP, and so it is perhaps unsurprising that it ranks positive NPP projections higher. Whilst the CbPM algorithms are portrayed as being ‘carbon-based’, our sensitivity analysis revealed that they are nonetheless predominantly driven by changes in CHL (Supplementary Information Fig. 7c,d), rather than changes in particulate backscatter, similar to previous studies (Song et al., 2024). Any divergence in the ranking between the two CbPM algorithms is most likely driven by how Westberry-CbPM is both depth and wavelength resolved to account for changing light properties as you move through the water column, as demonstrated by the higher ranking standard deviations for Westberry-CbPM (Fig. 4d), in comparison to the Behrenfeld-CbPM (Fig 4c). Finally the remaining algorithms, Lee-AbPM and Silsbe-CAFE, which more consistently rank negative projections highest, both respond strongly to changes in phytoplankton absorption (Supplementary Information Fig. S7e,f). In so doing, when determining phytoplankton productivity, these algorithms respond to the efficiency of light absorption, rather than the absolute quantity of photosynthetic pigments, which makes them better suited to capturing NPP responses to environmental variability (Marra et al., 2007).

During Primary Production Algorithm Round Robin exercises²²⁻²⁴ no single algorithm has been found to perform best at all times and locations. However, there is a general reduction in the root mean square difference between remote sensing NPP estimates and direct field measurements for the Lee-AbPM and Silsbe-CAFE algorithms (relative to the VGPM and CbPM algorithms), suggesting that they perform best overall (Supplementary Information Figure S8^{21,24-26}). Indeed, more recent studies that applied the Behrenfeld-VGPM, Westberry-CbPM and Lee-AbPM algorithms to OC-CCI data report similar findings where Lee-AbPM has the lowest RMSE (Wu et al., 2024). In addition, the Jackknife trend analysis on the 7 simulations (Supplementary Information Fig. S9) demonstrates that both the Eppley-VGPM and Behrenfeld-VGPM algorithms are strongly sensitive to the start or end dates of the time series (Supplementary Information Fig. S9a-d), with high coefficients of variation and even a switch in the dominant direction of NPP trends across the simulations. Although both CbPM algorithms had similarly high coefficients of variation across the globe (relative to the VGPM algorithms), they remain dominated by negative trends across all simulations, with some evidence of an increase in the magnitude of negative trends and the number of positive trends due to sensitivity of start or end dates (Supplementary Information Fig. S9e-h). The Lee-AbPM and Silsbe-CAFE algorithms displayed the most robust response in NPP trends to the jackknife simulations, with much lower coefficients of variation and no tangible increase in the number of positive trends, with only a slight increase in the magnitude of negative trends (Supplementary Information Fig. S9i-l). Those areas of the globe that display relatively higher coefficients of variation (e.g. the Southern Ocean) thus represent regions with reduced confidence in the magnitude of the predominantly negative trends, but not in their direction. Overall, this indicates that there are larger uncertainties for global NPP trends from the VGPM and CbPM algorithms, relative to the trends estimated from Lee-AbPM and Silsbe-CAFE. Together these points of consideration around NPP algorithm validation and trend sensitivity to the jackknife simulations suggest that the Lee-AbPM and Silsbe-CAFE algorithms are the most robust and therefore best suited for the implementation of the model ranking scheme. Since the implementation of the model ranking system according to these two algorithms most strongly favours models with negative Δ NPP, these results strongly support a greater likelihood of global NPP declines into the future.”

Furthermore, the methods section seems very hard to support the reproduction the study considering the information given. As I found the claim in this study quite important, the authors could expand and better describe the steps for reproducing the study. It could be beneficial for the authors, for example, to ask to a colleague to try to reproduce or understand all steps before resubmitting the manuscript if accepted. See the minor comments to more details.

R2.2 We have taken this comment on board and will be making all of the code for reproducing the figures available on GitHub with all of the input data stored in a Zenodo

repository. We have also edited the methods to make sections clearer and more detailed. Please see specific responses to your comments and also to Reviewer 3's comments below who had similar concerns over the level of details in the methods section.

Minor comments

"R" is for row or line

Summary paragraph

R20 Make it clear that the periods compared are not the same. Satellite trends are contemporary while Climate model trends are the difference from contemporary to 2100.

R2.3 We thank the reviewer for this suggestion and have amended the text as follows to make it clearer about the time periods.

Line 17: “Contemporary remote sensing records provide multi-decadal datasets³ that can assist in discriminating amongst divergent future NPP projections up to the end of the 21st century but have only been exploited at regional scales to date^{4,5}. Here we use global contemporary NPP trends (1998 - 2023) from six remote sensing algorithms to discriminate future projections of NPP across fifteen Earth system models.”

R23 Maybe “drive contemporary trends” is better explained as “based on the similarity of linear responses of NPP to SST, ETC, observed in the contemporary period”.

R2.4 We thank the reviewer for this suggestion and have amended the text as follows.

Line 22: “A model ranking scheme was developed based on the similarity of linear responses of NPP to changes in sea surface temperature, chlorophyll-*a* and the mixed layer.”

R38 Clarification is needed on how the areas are weighted and normalized. The variables are not shown in % per decade per m². Does it mean that larger areas have the same weight of small areas? Besides, why climate models and satellite estimations are not described as the same parameter most important for such study as Pg C per decade.

R2.5 We apologise for the confusion here on how we area weighted and normalised the NPP data. The NPP data were area weighted because the m² of a pixel at higher latitudes is less than that at lower latitudes, meaning any positive or negative trends at higher latitudes would be given a strong weighting when averaging spatially. For normalising the data we used mean-normalisation, where we took the average NPP over the time series and divided the whole time series by this number. We have added the following text to the methods to clarify these steps:

Line 401: “For spatially averaged biome annual mean NPP trends, either remote sensing or Earth system models, an ordinary least squares regression was applied to area weighted data normalised to the mean. Area weighting was determined as a function of latitude, such that remote sensing pixels (model grid points) at higher latitudes are smaller than remote sensing pixels (model grid points) at lower latitudes.”

When performing assessments of climate models it is most common to look at the overall change of the end of the future period (2081-2100) relative to the end of the historical period (1995-2014), which following correction for area weighting and correcting the units of per day gives us a units of Pg C per year. This approach allows us to remain consistent with analyses conducted over multiple CMIP cycles and IPCC assessment reports. It is not possible to report the changes from remote sensing in the same units, due to the short length of the time series. They are however comparable when looking at the % year⁻¹ units.

Main text

R68 Where do they best perform overall? The supplementary table S1 suggests that this performance is only assessed in tropical regions. Performance might differ depending on the site.

R2.6 We thank the reviewer for this suggestion and have expanded this section. We now include regions across the globe and have included some more recent studies. Please note that the RMSD numbers from Saba 2010 are essentially repeated in the Silsbe 2016 study, and the numbers from Ma 2014 are also repeated in the Tao 2017 study. As such the figure caption only refers to 4 studies now where the reader can access all of the information in the table. To make this information clearer we have plotted a heatmap where the colour represents RMSD, and we have made the algorithm with the lowest RMSD per region per study in bold with a thick box around it. Please see the figure below which will now become Supplementary Information Figure S8 in the manuscript.

Figure S9: Comparing differences between remote sensing NPP algorithms and direct field measurements. Root mean square differences (RMSD) between NPP estimated from algorithms (Eppley, 1972; Behrenfeld & Falkowski, 1997; Behrenfeld et al., 2005; Westberry et al., 2008; Lee et al., 2011; Silsbe et al., 2016) and

direct field measurements from the Bermuda Atlantic Time Series (BATS), Hawaii Oceanic Time Series (HOTS), Western Antarctic Peninsula (WAP), Ross Sea, Mediterranean Sea, coastal north east Atlantic (NEA), Black Sea, Arabian Sea, pelagic North Atlantic (NABE), Antarctic Polar Frontal Zone (APFZ), California coast (CALCOFI), Mediterranean Sea (DYFAMED), Scotia Sea (AMLR), Cariaco basin (CARIACO) (Saba et al., 2010; Silsbe et al., 2016; Tao et al., 2017; Wu et al., 2024). Cells with bold text and a black border represent the algorithm which had the lowest RMSD for the specific study. Please note that empty cells means that the algorithm was not implemented during the study.

Even with the additional studies and regions included we still believe that our original statement holds true, that overall Lee-AbPM and Silsbe-CAFE perform best in matchup comparisons relative to all the other algorithms.

R71 “mechanistic”: to my understanding the relationships are statistical and not exactly mechanistic. For example, the SST influence is not clearly distinguished between influence on growth rates or relationship with MLD. Could it be better formulated?

R2.7 Agreed, and following similar concerns from Reviewer 3 we have amended the text as follows:

Line 70: “In this work, we rank fifteen CMIP6 Earth system models according to their ability to capture the emergent contemporary relationships observed between NPP and environmental variables (sea surface temperature, chlorophyll-*a* and the mixed layer depth) in the 6 remote sensing algorithms.”

Contemporary global trends in remote sensing NPP and their drivers

R101 How can decadal trends be sensitive to shorter oscillations as ENSO? Pending clarification on methodology. Is a moving 10-year window average? How was the edges problem dealt with it? A decadal time series of decadal NPP of satellite and models could better clarify that also.

R2.8 We apologise if this section was not completely clear. We computed the trends of annual means, and then multiplied by 10 to get to decadal trends. There were no moving averages performed and thus we did not have to deal with any edge problems either. We used jackknife resampling to assess whether any climate events such as ENSO unduly influenced the final trend numbers we reported, in particular because 1998 (the start of our satellite record) is a known El Nino year.

R125 Are these regression over the decadal averages? Please, clarify it. If not, change to decadal averages as the study is focused on decadal trends.

R2.9 The multiple linear regressions are performed between annual means, we had multiplied them by 10 to make the trends per decade. We have now amended all trends to be per year, rather than per decade to avoid any further confusion.

R134 Two points should be considered here. First, the inclusion of new predictive variables can artificially increase the R². Showing the adjusted R² would better suit for the comparison. Second, clarify if and how the time auto-correlation of NPP was normalized for computing the p-value.

R2.10 We thank the reviewer for bringing our attention to this, we have now adjusted the methodology to account for autocorrelation and heteroskedasticity. At the same time we now report the adjusted R² values. Please see responses below where we have expanded the methods to properly document all of these changes. We have also amended the text in the main document as follows:

Line 108: “To statistically assess what drives local trends in NPP, we use multiple linear regressions (MLR) that account for unequal variance and autocorrelation. We used MLRs to link contemporary trends in NPP to a suite of environmental and biological drivers across all algorithms and jackknife trend simulations (see Methods).”

We have also added the following text in the methods to reflect the change in approach for how we determined the MLR coefficients, where we use jackknife resampling to run the MLR calculations on 7 different simulations.

Line 414: “Annual means of NPP, SST, CHL and MLD were first jackknife resampled to 80% of the time series, representing 7 different possible simulations, and then mean-normalised, i.e. the resampled time series was divided by its mean. Multiple ordinary least-squares linear regressions (MLR) were then performed using the Statsmodel package⁴⁶ using a Heteroskedasticity and Autocorrelation Consistent covariance estimator, where the the time lags for autocorrelation were calculated following Newey & West (1994), defined in Equation 6:

As a result of this change in analysis we found strong evidence of multicollinearity between MLD_{min} and MLD_{max}. To circumvent this issue we have switched to using annual mean MLD instead, referred to as MLD in the text.

Assessing Earth system model projections of NPP

R169 “Over the historical period” Why use the historical period? As showed in Figure 1, the decadal trends vary among the periods: historical, contemporary, and future. Since this variability happens and the authors want to compare with the contemporary satellite observations, it seems more plausible to use the climate models under the contemporary period. Furthermore, it is not clear if the regression is on decadal averages. Considering the comparison with Figure S6, it would better suit to show the adjusted R² and how the time autocorrelation was normalized for estimating the p-value.

R2.11 We analyse climate models over the longer term historical period to be consistent with prior modelling exercises. The reasoning for this is that the reliability models can only be assessed by observations of the past and present. This means that models are assessed against criteria that are not necessarily informative in terms of the quality of model projections of future climate change. The emergent constraint approach attempts to address this problem by identifying robust, physically interpretable relationships between Earth system feedback behaviours on well-observed timescales (Eyring et al., 2019, Nature). The fidelity of an emergent constraint, thus, depends on the correlation of the relationship and the uncertainty of the observations (Terharr et al., 2023, Science Advances).

As mentioned above we perform the multiple linear regressions on the annual means, we also now account for autocorrelation and report the adjusted R^2 values.

R194 Would be beneficial to explain what a low value and high value means.

R2.12 Please note that this section has been revised following the switch from MLDmax/min to only MLD. Additionally please note that we no longer refer to high or low values, please revised sentence below:

Line 177: “MLD coefficients show a large discrepancy in their zonal distribution relative to remote sensing (Fig. 3f) with coefficients that are broadly positive and more similar to each other at high latitudes but larger and more diverse (spanning both positive and negative coefficients) at lower latitudes.”

R197 “10TH and 90TH percentile to remove extreme outliers”. The authors are removing 20% of the data, which does not sound as extreme outliers. Reduce it to 5% (2.5% and 97.5%) or give a plausible explanation of this wide range of removal.

R2.13 We thank the reviewer for raising this issue around what is considered an outlier and have made the following changes to the manuscript. We first examined whether we could use Z-scores to classify outliers, but this requires that the data be normally distributed, where our MLR coefficient distributions are not. Then we looked at the use of IQR limits, nominally known as the fence test, because this has no assumption of normality and can be more robust to outliers. With this in mind we now classify outliers using $IQR \pm IQR \times 3$, i.e. outliers are values greater than $IQR + IQR \times 3$ (and vice versa). This IQR factor is usually set to 1.5, but we have opted to be more relaxed in this regard so as not to exclude the majority of the data. Below are 2 examples demonstrating this.

If we set the IQR factor to 1.5, the most common use of this outlier method, we lose most of the spatial variability in the MLR coefficients. However if we set it at 3 then we still retain the majority of the spatial variability in the MLR coefficients. We find that this approach for classifying outliers using a standardised method, as opposed to just removing upper/lower quantiles, is best suited for our data analysis. Please see the new final revised Figure 3 below.

“Figure 3: Comparing the spatial variability of the dominant multiple linear regression coefficients between remote sensing and Earth system models. Zonal averages \pm standard deviations of the multiple linear regression coefficients for (a,d) sea surface temperature (SST), (b,e) chlorophyll-a concentrations (CHL), and (c,f) mixed layer depth (MLD) for the (a-c) Eppley-VGPM, Behrenfeld-VGPM, Behrenfeld-CbPM, Westberry-CbPM, Lee-AbPM and Silsbe-CAFE NPP algorithms and (e-f) the ensemble of CMIP6 Earth system models. Only pixels/grid points where the multiple linear regression analysis was significant are included in the zonal averages. The shaded region in panels d-f represents the range of coefficients as estimated from the remote sensing algorithms zonal averages (panels a-c).”

We have also amended the relevant text in the main document as follows.

Line 174: “EMD metrics were calculated per jackknife simulation and per biome²⁷, with MLR coefficients restricted using the interquartile range fence test (see Methods) to remove extreme outliers in each biome.”

We also added additional text in the methods:

Line 436: “The MLR coefficient values for both the remote sensing and models were restricted using the interquartile range (IQR) fence test, $IQR \pm IQR \times 3$, to remove any extreme outliers.”

We have also added an additional figure that focuses on the across simulation variability to demonstrate that the spatial variability is greater than the variability between simulations.

“Figure S4: Comparing the Jackknife simulation variability of the multiple linear regression coefficients of the remote sensing algorithms. Jackknife simulation averages \pm standard deviations of the multiple linear regression coefficients for (a) sea surface temperature (SST), (b) chlorophyll-a concentrations (CHL), and (c) mixed layer depth (MLD) for the Eppley-VGPM, Behrenfeld-VGPM, Behrenfeld-CbPM, Westberry-CbPM, Lee-AbPM and Silsbe-CAFE NPP algorithms.”

We have also added this additional text to the manuscript:

Line 148: “Consistent across all algorithms the spatial standard deviations (Fig. 3a-c) were always substantially greater than the standard deviations of the jackknife simulation (Supplementary Information Fig. S4), confirming that regional variability plays a more significant role in the proposed ranking scheme.”

R211 Please, better clarify the explanation of what is shown in figure S8. Are they the z-score mean and standard deviation of what distribution? I found this part difficult to understand.

R2.14 We apologise if this part was unclear. As we calculated the EMD for 4 driver coefficients (SST, CHL, MLD_{max} , MLD_{min}) per CMIP6 model we generated a driver mean and standard deviation. Next we calculated the Z-score for both driver EMD mean and driver EMD standard deviation, before averaging these 2 individual Z-scores to give us 1 final value per CMIP6 model. This exercise was to try and find the best way to summarise the level of information so that we could rank the models. We state on line 215 how we generated this EMD mean and standard deviation, following the revision to only include 3 variables as part of our analysis.

Line 202: “Next, we average the EMD values across all three variables, SST, CHL and MLD, to generate a single EMD mean and standard deviation for each Earth system model per remote sensing algorithm jackknife simulation.”

We have revised the sentence for referencing Figure S8 to hopefully make this clearer, however please note that the original Figure S8 has been removed, with the information now included in the new Figure 4.

Line 204: “For each remote sensing algorithm we then rank the CMIP6 models using Z-scores that incorporate both the EMD mean and standard deviation.”

R217 The dark grey in Figure 4 legend (Rank 2 in Lee-abpm) seems to be missing. Besides, one bar is too small that the colour is not visible, please, fix it.

R2.15 We thank the reviewer for finding this error in the figure legend, please note that we have now amended Figure 4 (see below). For the bar which is too small, this is the ACCESS ESM which has a ΔNPP value close to 0, so unfortunately there is no easy way to make it visible. However since we have amended Figure 4 to reflect our new approach this problem no longer persists.

Are future declines in NPP underestimated?

R250 Change “CHL” to the standard used in the text.

R2.16 We are confused as to why we would change CHL in this paragraph here. We defined the acronym CHL to reflect either chlorophyll-a as measured by remote sensing or chlorophyll-a in the Earth system models. This is the standard we use throughout the manuscript to make this distinction about how we performed our ranking exercise. So we believe it is ok to continue using CHL in the text here.

Material and Methods

Remote Sensing Net Primary Production

R311 Please, describe at least the simplified versions of each NPP equation applied in the satellite observations. Although some sort of normalised response is shown in Figure S1, it would be more much beneficial for the reader to assess the equations that the author employed. This could be easily done and very important as they are used to rank the climate models.

R2.17 We thank the reviewer for this comment and have amended the beginning part of the methods as follows to include simplified equations. Please note, it is not possible to include every equation required for all derivations as this would substantially increase the length of the manuscript and we do not feel it is necessary as they are properly documented in their original publications.

Line 324: “Net primary production (NPP; mg C m⁻² d⁻¹) was calculated using the following algorithms, the ‘vertically generalised production model’s (Eppley-VGPM¹⁶ & Behrenfeld-VGPM¹⁷; Equation 1), which relies on the relationship between chlorophyll *a* and temperature derived growth rates; the ‘carbon-based production model’s (Behrenfeld-CbPM¹⁸; Equation 2 & Westberry-CbPM¹⁹; Equation 3), which uses backscatter derived phytoplankton carbon as a biomass indicator and physiology derived as variability in the chlorophyll *a* to carbon ratio; the ‘absorption-based production model’ (Lee-AbPM²⁰; Equation 4), which does not make any assumptions on either chlorophyll *a* or backscatter as biomass proxies but relies on absorption characteristics to infer phytoplankton photosynthetic efficiency; and the ‘carbon, absorption, and fluorescence euphotic’ resolving model (Silsbe-CAFE²¹; Equation 5), which derives NPP as a function of energy absorption and efficiency.

Equation 1:
$$NPP = Chl \times f(T) \times DL \times f(PAR) \times Z_{eu}$$

Where Chl is chlorophyll-*a* concentration, $f(T)$ is a temperature based growth function (exponential for Eppley-VGPM and a 4th-order polynomial for Behrenfeld-VGPM), DL is day length in hours, $f(PAR)$ is the parameterized light term for the ratio of realised NPP to maximum potential NPP and Z_{eu} is the depth of the euphotic zone.

Equation 2:
$$NPP = C_{ph} \times \mu \times I_g$$

Where C_{ph} is phytoplankton carbon derived from an empirical relationship with particulate backscatter at 443 nm ($b_{bp}(\lambda_{443})$), μ (μ_{max}) is the growth rate and I_g is the growth irradiance term.

Equation 3:
$$NPP = \int_0^{Z_{NO_3}} \mu \times I_g dz$$

Where Z_{NO_3} is the depth of nitracline, defined as the depth at which nitrate + nitrite exceed 0.5 μ M, and dz is the depth.

Equation 4:
$$NPP = a_{ph}(\lambda_{443}) \times K_d(\lambda_{490}) \times PAR$$

Where $a_{ph}(\lambda_{443})$ is phytoplankton specific absorption at 443 nm, $K_d(\lambda_{490})$ is the light attenuation coefficient at 490 nm and PAR is the daily available photosynthetic radiation.

Equation 5:
$$NPP = f(T) \times f(PAR) \times Z_{eu}$$

Where Q_{PAR} is energy absorption, $\phi_{\mu}^{\square\square\square}$ is efficiency at which the absorbed energy is converted into carbon biomass, E_k is the light saturation parameter and E is daily PAR. For more details on all equations please refer to their specific publications.”

R320 Better describe the satellite products used. For example, clearly state where each variable used in the manuscript came from which product, the time frequency of each product, the uncertainty when available, and give the proper citation of these products (each website product often has a section describing how to cite their data when using them, follow those guidelines). It should also address some known limitations that might impact this work: i) given the optical observations are lacking in high latitudes during the winter, how the NPP was corrected given that there is no data during periods of very low NPP in this region; ii) the MLD thresholds used is 0.03 kg m⁻³, which may not be the same threshold used in some of the climate models (0.01 kg m⁻³), state this limitation and any foreseen potential impact.

R2.18 We have amended the text to include some more details about time frequency and spatial resolution of the products to the materials and methods. To the best of our knowledge we have cited each product correctly, where certain products do not contain any specific information on their website of how to cite we have included as many details as possible in the methods, data availability statement and acknowledgements. We have added the following text to the acknowledgements for GlobColour:

“GlobColour data (<http://globcolour.info>) used in this study has been developed, validated, and distributed by ACRI-ST, France.”

And here is the modified text of the methods:

Line 391: “The algorithms were applied to ocean colour remote sensing data from the European Space Agency Ocean Colour Climate Change Initiative (OC-CCI) data product (8-day, version 6.0³) from 1998 to 2023, which was regridded to 25 km using bilinear interpolation. Photosynthetically active radiation (PAR; mol photons m⁻² d⁻¹) was taken from the merged GLOBColour product (<http://globcolour.info>) at 25 km 8-day resolution. For VGPM sea surface temperature (SST; °C) was taken from the Group for High Resolution Sea Surface Temperature (GHRSSST; <https://www.ghrsst.org/>), which was regridded to 25 km as above. For CbPM and CAFE the mixed layer depth (MLD; m) was taken from the Hadley EN 4.2.2 gridded temperature and salinity profiles³⁵, which were first regridded to 25 km as above and resampled to 8-days, then converted to density using the Gibbs Seawater TEOS-10 python package and the MLD derived from a density criterion of 0.03 kg m⁻³ and reference depth of 10 m³⁶. Full explanation of the VGPM, CbPM and CAFE NPP calculations is provided by Ryan-Keogh et al.³⁷, with data publicly available³⁸. For AbPM we used the OC-CCI $a_{\text{ph}}(\lambda 443; \text{m}^{-1})$ and $K_d(\lambda 490) (\text{m}^{-1})$, in combination with the GLOBColour PAR, with data publicly available here³⁹.”

We do wish to draw your attention to that information on how we processed 5 out of the 6 algorithms is already published in ESSD, which is directly cited here in the text. To

avoid repetition we are opting not to include any further details than what is already provided, where the readers can use the ESSD reference and data availability statements to access all the information they may require. Duplicating this information did not seem to be appropriate here.

To the best of our knowledge the data products which provide uncertainties are OC-CCI (except for $b_{bp}(\lambda 443)$), GHRSSST (SST analysis error) and GlobColour (PAR sensor averaging error). However, as stated in Song et al. (2024; Remote Sensing of Environment), because of NPP algorithm complexity it is challenging to derive an analytical formula following error-propagation theory (Melin, 2019; IOCCG Uncertainties in Ocean Colour Remote Sensing) to estimate uncertainties in NPP algorithms in relation to the uncertainties of each individual input. Until such a time as this analytical formula can be developed then we are unable to provide uncertainties related to our NPP estimates. Furthermore, this only covers the uncertainties of the remote sensing data products and how that impacts the derivation of NPP with the different algorithms. A true estimate of NPP uncertainty would require a global database of NPP measurements from which we would assess the validity of match ups. This is something which the round robin exercises are attempting to do, but currently there are not enough measurements across the global ocean for a full assessment as there is for example when validating chlorophyll-a concentrations. Addressing this is beyond the scope of this paper, but we do agree it is an important issue for future studies.

Line 316: “Future assessments should not only consider the uncertainties inherent to remote sensing algorithms (Supplementary Information Fig. S8), despite the complexity in deriving them (Song et al., 2024), but should also expand on the Round Robin intercomparison exercises (Supplementary Information Fig. S8) as more in situ data becomes available. Furthermore, future model assessments should consider using additional parameters in combination with those proposed here, such as the resource limitation diagnostics in Earth system models (e.g. iron limitation, light limitation etc), which could be used to assess ongoing changes in the Southern Ocean³³ and the equatorial Pacific³⁴.”

We acknowledge your concern about NPP estimates at high latitudes where data is missing during the winter months due to the cloud cover, where we could be underestimating the total annual mean NPP. As such we have added a small section to the methods to acknowledge this caveat. Please see response below (**R2.22**) where we discuss further your concerns around the high latitudes.

Line 408: “Please note that the remote sensing annual means of high latitudes may be potentially underestimated due to the presence of cloud cover preventing the retrieval of data.”

For your final concerns regarding the issues of different criterion being used for MLD determination we are unfortunately constrained by utilising the only information which is provided to us when we download data from the Earth System Grid Federation. All 15

CMIP6 models state that MLD is derived using density, but no models provide any information on what this criterion is.

Earth System Model Selection and Download

R341 Time frequency of the models?

R2.19 We thank the reviewer for this suggestion and have added additional clarity in the text as follows:

Line 385: “All data variables were regridded on a regular $1^\circ \times 1^\circ$ grid using the bilinear interpolation of Climate Data Operators⁴⁰, and were resampled from from a monthly resolution to annual means.”

Calculating Decadal Trends

R348 Clearly separate what is the method for the satellite trends and for the climate models trends.

R2.20 We thank the reviewer for this suggestion and have added additional clarity in the text as follows:

Line 391: “Trends of remote sensing annual mean NPP were calculated by first excluding any pixel whose time series had less than 50% of the data available.”

Line 403: “For spatially averaged biome annual mean NPP trends, either remote sensing or Earth system models, an ordinary least squares regression was applied to data normalised to the mean.”

R348 Describe in more details the method for estimating the trends. Is it a 10-year moving window average? How was the edge problem of the time series resolved? What is the period interval?

R2.21 We performed the trend analysis on annual means from the 26 year dataset, and then multiplied the slope by 10 to present the results as per decade. We did not use a 10-year moving window average, and as such did not have to resolve any edge problems from smoothing. However to avoid any further confusion we no longer convert any trends to per decade, but instead keep all trends as per year.

R348 What are the impact of removing the pixels in the high latitude regions on the estimations of global NPP? In satellite observations, it is often lacking data in these regions because of the long dark winter and cloudy conditions. See figure S5 and it is noticed the white areas in these parts. Given that high latitude regions are expected to have a substantial increase in NPP mainly driven by reduction of sea ice among others (some examples: <https://doi.org/10.1016/j.pocean.2015.05.002>,

<https://doi.org/10.1038/s41467-020-20470-z>), how not including these positive anomalies could affect the global decadal trend estimate in this study?

R2.22 We thank the reviewer for raising this point about the issue of missing pixels at high latitudes, particularly during winter. We would first like to note that during long dark winters I believe it is safe to assume that phytoplankton productivity is minimal and therefore we are not likely to be missing any potentially large blooms. Secondly, unlike the study you have referenced above, which used only 1 satellite sensor with no gap filling employed, we utilise the OC-CCI data product, which merges all available satellite sensor missions, therefore increasing the potential number of pixels measured. Additionally, we also employ a very common gap filling exercise, where missing pixels were filled using a linear interpolation scheme in sequential steps of longitude, latitude and time (Racault et al., 2014) using a three-point window. If one of the points bordering the gap along the indicated axis was invalid, it was omitted from the calculation, whilst if two surrounding points were invalid, then the gap was not filled. This is all documented in the ESSD paper for the NPP data product (Ryan-Keogh et al., 2023). With all of this we would conclude that our NPP data has more pixels of valid data than the study you have referenced.

One final point to mention, is that we calculate the EMD values per biome and then scale those values based upon that biome's relative proportion of the global ocean. As such, the relatively small ice biome regions when compared to the globe, particularly in the north, make a small contribution to the final biome weighted Z scores. So again we would conclude that even with some potentially missing data at high latitudes, their influence on the final ranking exercise will be minimal.

R350 What normal distribution test? (e.g., Shapiro-wilk?)

R2.23 We ran the D'Agostino-Pearson normal distribution test. We have amended the methods to:

Line 392: "Before linear regressions were performed, the data were first tested for a normal distribution using the D'Agostino-Pearson test in the SciPy python package⁴¹."

R352 How can more than 50% of a distribution be the outliers? If they are a normal distribution, it would not be impossible to more than 50% of the data being outliers?

R2.24 The Huber-Regressor function specifically has a function to control the robustness, i.e. the potential number of samples which can be classified as outliers. Huber (1981) recommends this is set to 1.35 (see answer below). However there are some cases where maintaining a pixel's time series robustness, using the prescribed parameters from Huber,

would mean more than 50% of the 26 years would be classified as outliers. As it is recommended to keep this robustness factor constant, rather than estimating from the data (Huber, 1981), we kept it as 1.35 for all of our analyses. So we decided to exclude these pixels to maintain both criteria for significant trends. See next response (R2.25).

R352 What is ϵ used for and why 1.35?

R2.25 ϵ is the parameter which controls the number of samples which should be classified as outliers. Huber (1981) recommended that it is set at 1.35 to get as much robustness as possible while retaining 95% statistical efficiency for normally distributed data. We have added the following text to the methods to clarify this more clearly:

Line 394: “If the data were normally distributed, then linear regressions were performed using the Sci-Kit⁴² Huber-Regressor, where ϵ , the parameter to control the amount of robustness (i.e. the number of outliers), was set to value of 1.35. This value is to ensure maximum robustness whilst maintaining 95% statistical efficiency (Huber, 1981). If a pixel had less than 50% of the time series following outlier removal then no further tests were performed.”

R354 Could the different methods of estimating trends generate different slopes as an artificial outcome from the method given if it employed both in normal distributions?

R2.26 The different methods were used following the test for normal distribution. Please see the text below where we state which regression was used for a normal distribution (Huber-Regressor), and which regression was used for a non-normal distribution (Mann-Kendall). Mixing these up does not seem like a good strategy as they are designed to be used as implemented.

Line 394: “If the data were normally distributed, then linear regressions were performed using the Sci-Kit⁴² Huber-Regressor, where ϵ , the parameter to control the amount of robustness (i.e. the number of outliers), was set to value of 1.35. This value is to ensure maximum robustness whilst maintaining 95% statistical efficiency (Huber, 1981). If a pixel had less than 50% of the time series following outlier removal then no further tests were performed. If the data were not normally distributed, then linear regressions were performed using the non-parametric Mann-Kendall Test⁴⁴.”

Multiple Linear Regression and Earth Movers Distance Analysis

R364 Why annual means? The interannual modelled response would reflect the response in a decadal time scale? Is there any limitation on that?

R2.27 We use annual means for the multiple linear regression analysis in a consistent manner between the products. Indeed, this would reflect the decadal scale trends. We do not see why this would indicate any limitations. However to avoid any further confusion we no longer convert any trends to per decade, but instead keep all trends as per year.

R364 Why the variance was not normalized?

R2.28 Thanks for this comment. We have amended the method for the multiple linear regression analyses to now account directly for heteroscedasticity (i.e. unequal variance) and autocorrelation consistent standard errors. Please see responses to R2.29 for more details as the change in methodology applies to both comments.

R365 The autocorrelation was checked but what was done about it? It is difficult to not find autocorrelation in annual means of the ocean given the oceans long memory, especially for SST. If auto correlation was found, how it was corrected in the regressions? Mention the autocorrelation method instead of the tool, or both if necessary.

R2.29 We agree that more detail is needed. We have amended the method for the multiple linear regression analyses to now account directly for heteroscedasticity (i.e. unequal variance) and autocorrelation consistent standard errors. This is all performed within the Statsmodel package where we defined the maximum lags for autocorrelation using the Newey and West (1994) rule of thumb.

Line 415: “Annual means of NPP, SST, CHL and MLD were first jackknife resampled to 80% of the time series, representing 7 different possible simulations, and then mean-normalised, i.e. the resampled time series was divided by its mean. Multiple ordinary least-squares linear regressions (MLR) were then performed using the Statsmodel package⁴⁶ using a Heteroskedasticity and Autocorrelation Consistent covariance estimator, where the time lags for autocorrelation were calculated following Newey & West (1994), defined in Equation 6:

Equation 6:
$$lag = \left\lceil 4 \times \left(\frac{T}{100} \right)^{2/9} \right\rceil$$

where T is the length of the time series, which in this case is 26 years for remote sensing and 165 years for the Earth system models. No MLR was performed for a remote sensing pixel or model grid point if any variable was missing data from any year of the time series or if the variance for any of the drivers was ~0. MLR Coefficients for each prospective driver were excluded from further analysis if either the remote sensing pixel or model grid point were not significant (p>0.05).”

Following this change in our statistical approach, we found evidence of multicollinearity between MLDmin and MLDmax. We opted to not include any correction schemes for this, but rather adjusted our analysis to focus on annual mean MLD instead, which avoided the autocorrelation issue.

R366 Name the method for the regressions.

R2.30 We have amended the text as follows:

Line 417: “Multiple linear regressions were then performed using the ordinary least squares function from the Statsmodel package⁴⁶ using a Heteroskedasticity and Autocorrelation Consistent covariance estimator”

R369 Significance test and correction on autocorrelation if necessary?

R2.31 We have amended the method for the multiple linear regression analyses to now account directly for heteroscedasticity (i.e. unequal variance) and autocorrelation consistent standard errors. Please see response above (R2.29) where we have amended all the details of the multiple linear regression analyses. In the text we defined significance at the 95% significance level ($p < 0.05$).

Line 425: “No MLR was performed for a remote sensing pixel or model grid point if any variable was missing data from any year of the time series or if the variance for any of the drivers was ~ 0 . MLR Coefficients for each prospective driver were excluded from further analysis if either the remote sensing pixel or model grid point were not significant ($p > 0.05$).”

R374 Correct the equation and describe below the variable l_1 .

R2.32 We thank the reviewer for this suggestion and have amended the text as follows:

Line 429: “Comparisons between the observational data products and model coefficients were performed using the Earth mover’s distance (EMD) metric³², also known as the Wasserstein distance in mathematics⁴⁷ and Mallow’s distance in statistics⁴⁸, defined here in Equation 7:

Equation 7:
$$l_1(u, v) = \int_{-\infty}^{+\infty} |u - v|$$

where l_1 is the first EMD, u and v are the respective distributions of the MLR coefficients from remote sensing and Earth system models and U and V are the respective cumulative distance functions of u and v .”

R379 Why remove 20% of the data as outliers instead of 5%? I don’t consider 5% as a standard value but 20% looks excessive and demands an explanation.

R2.32 We thank the reviewer for raising this concern, as this was an oversight on us. We have since amended the outlier detection scheme to follow more commonly used methods. See response to R2.13 for more details. We have amended the text in the methods to reflect this change.

Line 437: “The MLR coefficient values for both the remote sensing and models were restricted using the interquartile range (IQR) fence test, $IQR \pm IQR \times 3$, to remove any extreme outliers.”

R387 Describe better this equation. For example, x is what value of the model, etc.

R2.33 We thank the reviewer for this suggestion and have added the following text to the methods:

Line 422: “To rank the models the Z-score, also known as standard score, was calculated using Equation 8:

Equation 8:

$$z = \frac{x - \mu}{\sigma}$$

where x is either the model’s EMD mean (or standard deviation), μ is the model ensemble mean of either the EMD mean (or standard deviation) and σ is the model ensemble standard deviation of either the EMD mean (or standard deviation). The final Z-scores, determined from both the EMD mean and the EMD standard deviation, were then generated by combining with equal weighting (i.e. the Z-scores were averaged together.”

Reviewer #3 (Remarks to the Author):

COMMSENV-24-1567-T Review signed by Benoit Pasquier

manuscript title: Global decline in net primary production underestimated by climate models

The ocean's biological primary productivity sustains global marine ecosystems and is tightly linked to the global carbon cycle and climate. Accurate projections of the ocean's productivity for the next century are thus critically important given the dramatic changes expected from our rapidly warming climate. However, there is a major issue with the current state of the science. Climate models disagree on the magnitude and even the sign of predictions of net primary production (NPP) over the next century.

A likely related issue is that remote-sensing estimates also disagree on the magnitude and sign of the NPP trend of the past few decades. To tackle these issues, the authors compare and rank the CMIP6 climate models according to their ability to match historical NPP estimates from a suite of remote-sensing models.

An important feature of the climate-model rankings is that it is based on the sensitivities of NPP estimates to environmental variables.

The major claim of this paper, which is clearly laid out in the title, is that CMIP6 models underestimate the future NPP decline. This claim is mainly supported by two arguments: (i) the rankings of climate models, of which those predicting strong future NPP declines tend to be ranked better, and (ii) the mismatch between the low SST sensitivities of climate models and the high SST sensitivities of remote-sensing algorithms, which suggests that more accurate climate models would predict even stronger NPP declines in the future than what they currently predict.

To the best of my knowledge, the claim and the arguments that support it are novel. The paper will be of interest to many, including in the fields of oceanography, biogeochemistry, climate-modelling, as well as policymakers (if they understand the conclusions).

I commend the authors for good work in that the manuscript is clear, short, and well-structured, and the figures and supplementary material are generally adequate and help to understand the main story.

However, I think many minor things could be improved.

Hence, in my opinion, this work is worthy of publication in *Communications Earth & Environment* after some revisions.

The most important revisions the authors should consider are the following in my opinion:

1. General presentation improvement to emphasize the science over the statistics (some details below)

2. Improve and merge Fig 4 into Fig 1 (Fig 4 is the central Figure but can be greatly improved)

3. Add a paragraph of discussion of the caveats of the method, which is missing. I am unsure what the biggest caveats are, if any, but one issue that I think needs at least a sentence is that the central claim hinges on a comparison of CMIP6 climate models with a suite of remote-sensing models that are afflicted by large uncertainties themselves. While the authors acknowledge and discuss these uncertainties, they don't discuss their effects on the conclusions drawn. In an ideal world, one would directly compare the climate models' NPP to observations of NPP and avoid the need for remote-sensing algorithms altogether. However, remote-sensing NPP estimates are the best tool we currently have for estimating NPP with global coverage from variables observed by satellites. To me, this begs the question: How would systematic bias in the suite of remote-sensing models used in the authors' analysis affect their conclusions? Another set of caveats may lie in the choice of environmental drivers, which seems arbitrary to some extent. What other drivers could have been included? What important driver could be missing, if any? Or is there reasonable confidence that the SST, CHL, and MLD set is optimal?

R3.1 We thank the reviewer for raising this concern of systematic bias in the remote-sensing models used. We attempted partially to address this with original Figure S1 (now Figure S2) which determines which input variable is the primary determinant of NPP. However we are aware that for each input variable there is a bias in comparison to in situ measured variables. Similar to reviewer 2 (**R2.18**) who also raised concerns around uncertainties we have added the following statement to the end of the manuscript.

Line 316: “Future assessments should not only consider the uncertainties inherent to remote sensing algorithms (Supplementary Information Fig. S8), despite the complexity in deriving them (Song et al., 2024), but should also expand on the Round Robin intercomparison exercises (Supplementary Information Fig. S8) as more in situ data becomes available. Furthermore, future model assessments should consider using additional parameters in combination with those proposed here, such as the resource limitation diagnostics in Earth system models (e.g. iron limitation, light limitation etc), which could be used to assess ongoing changes in the Southern Ocean³³ and the equatorial Pacific³⁴.”

For your concern around the choice of drivers again we are aware that we are currently limited in scope, but this is due to 1) the requirement of needing an observable metric over the same time period and 2) the requirement that the CMIP6 outputs have an

analogue with which we can compare to. In future assessments we advocate for the use of resource limitation diagnostics, such as iron limitation, but with the current CMIP6 models we are using only 7 of them make this output regularly available, and even then this metric is only available for the surface rather than over the water column. We are aware that as part of CMIP7 there is a community review to highlight which model outputs are the most important for assessment, and so future assessments of ESM's ability to predict NPP trends will be able to take advantage of this.

We are also aware that each NPP algorithm has implicit biases and assumptions around how they derive NPP, which we raise as a concern with the sentences below:

Line 51: "Trends in marine NPP estimated from remote sensing however also vary considerably depending on the time period, algorithm implemented, and data product being used¹²⁻¹⁵. Some of the sensitivities to time period and data product are addressed by the generation of a coherent multi-sensor satellite record spanning 1998-2023 that merges all available single-sensor satellite missions with substantially reduced inter-sensor biases³. Nonetheless, intrinsic differences in remote sensing trends are still apparent in the range of algorithms available for quantifying NPP rates."

By implementing 6 different algorithms we are in essence accounting for these biases when algorithms behave similarly in the ranking exercises. In an ideal situation we would perform this ranking exercise only with algorithms which are fully validated with in situ measurements, something which we have advocated for in future assessment exercises:

Line 316: "Future assessments should not only consider the uncertainties inherent to remote sensing algorithms (Supplementary Information Fig. S8), despite the complexity in deriving them (Song et al., 2024), but should also expand on the Round Robin intercomparison exercises (Supplementary Information Fig. S9) as more in situ data becomes available."

4. If possible, the manuscript would be greatly improved by some brief discussion on what could actually be done to improve climate models (and remote-sensing algorithms) to achieve better consensus in NPP estimates and projections. Maybe these papers could guide this discussion:

- Henson et al. (2022; <https://www.nature.com/articles/s41561-022-00927-0>)
- Boyd (2015; <https://www.frontiersin.org/journals/marine-science/articles/10.3389/fmars.2015.00077/full>)

R3.2 We thank the reviewer for raising this concern however we had assumed that this point we raised was our recommendation for how to improve climate models, based upon the analysis we performed.

Line 294: "Accordingly, an improved reproduction of contemporary trends in NPP from Earth system models suggests NPP needs to become more sensitive to SST increases and less sensitive to CHL increases."

We thank you for suggesting the two studies about how we can improve the performance of Earth system models. We have revised the manuscript to include the following statements:

Line 318: “Future assessments should not only consider the uncertainties inherent to remote sensing algorithms (Supplementary Information Fig. S8), despite the complexity in deriving them (Song et al., 2024), but should also expand on the Round Robin intercomparison exercises (Supplementary Information Fig. S9) as more in situ data becomes available. Furthermore, future model assessments should consider using additional parameters in combination with those proposed here, such as the resource limitation diagnostics in Earth system models (e.g. iron limitation, light limitation etc), which could be used to assess ongoing changes in the Southern Ocean³³ and the equatorial Pacific³⁴. Such approaches would deliver greater confidence in the mechanistic representation of NPP in Earth system models necessary to project associated impacts on marine ecosystems and biogeochemical cycles.”

5. Improvements to the Methods section (particularly the part on Multiple Linear Regressions and Earth Mover's Distance).

We thank the reviewer for their time and conscientious suggestions. We respond below to all of their issues.

Below I detail all my suggestions (except for points 3. and 4. above) in order of appearance in the paper, including much more minor issues.

- L14: **Some would contend that NPP is not a "major" flux** when compared to other fluxes and I think the most important part here is that NPP sustains ecosystems anyway, so what about starting with it, e.g., "... (NPP) supports critical ecosystem services and is important for the carbon cycle".

R3.3 We thank the reviewer for this suggestion and have amended the text as follows:

“Marine net primary production (NPP) supports critical ecosystem services and is important for the carbon cycle¹.”

As an aside, the NPP flux is around 50 Pg C per year and thus represents a carbon flux that is more than 20-times as large as air-sea CO₂ fluxes and perhaps 5 times as large as export production fluxes. We are happy to amend the text, but could not think of a larger C flux in the system.

- L22: I think I understand that the authors want to hint that they don't just use yearly-maximum MLDs, but everything is "seasonal" by nature in the ocean. What about removing "seasonal" and just use "mixed layer".

R3.4 We thank the reviewer for this suggestion and have removed the word seasonal.

- L23–25: It seems obvious to me that a "model ranking scheme" is "able to sort models" and I don't think it is useful to say here in the summary that it can reduce across-model variance. What about something simpler and punchier like: "These rankings suggest that a future decline in global NPP is more likely than not and that this decline is currently underestimated by all climate models."

R3.5 We thank the reviewer for this suggestion and have amended the text as follows:

Line 23: "These rankings suggest that a future decline in global NPP is more likely than presently assessed."

- L25–28: This sentence is a little unclear to me. What about splitting it into something like: "In addition, we find that models tend to statistically underestimate the NPP decline driven by sea surface temperature (SST) warming. This suggests that more accurate climate models that capture this higher SST sensitivity would predict even greater NPP declines in our warmer future climate." (I would remove the redundant "with important consequences for the marine ecosystems" since NPP was already said to support ecosystems in the first sentence of the summary paragraph.)

R3.6 We thank the reviewer for this suggestion and have amended the text as follows:

Line 25: "Additionally, we find that models tend to statistically underestimate NPP decline associated with ocean warming. If future climate models were able to capture this higher SST sensitivity, even greater NPP declines in a warmer future climate would result."

- L31–36: I don't think it is entirely correct to say NPP supports ecosystem services by sustaining biodiversity. In addition, I don't think that the role that NPP plays in the carbon cycle is important in this paragraph, which is about the importance of NPP for ecosystems and its uncertain future. So what about starting with that instead, with something along the lines of: "Marine NPP by phytoplankton sustains biodiversity and is essential to ocean ecosystems, but its future is uncertain." And then dive into the details of this uncertainty and the urgency of dealing with it.

R3.7 We thank the reviewer for this suggestion and have amended the text as follows:

Line 30: “Marine net primary production (NPP) by phytoplankton sustains biodiversity and is essential to global ocean ecosystems, but its future is uncertain¹.”

- L41: I'm not sure that calling NPP a "boundary condition" is correct, but more importantly, I don't think it helps to understand this sentence anyway, so what about: "(...) utilise NPP projections from only two climate models (...)", which is a bit shorter, too?

R3.8 We thank the reviewer for this suggestion and have amended the text as follows:

Line 39: “Furthermore, upper trophic level models that assess future responses of fisheries typically subsample NPP projections from at the ‘high’ and ‘low’ extremes of available projections^{8,9}.”

- L46–49: This sentence is a bit long and contains redundancies, and although it has been used elsewhere, I don't think "emergent constraint" is correct or useful here (the changes and relationships are emergent, but the constraints are not, even if using some relationship as a constraint is novel). What about something like: "Remote-sensing estimates of NPP over the contemporary period (1998-2023) provide global constraints for Earth system models. In addition, emergent relationships between changes in NPP and concomitant changes in ocean environmental variables over the contemporary period provide further constraints for Earth system models."

R3.9 We thank the reviewer for this suggestion and have amended the text as follows:

Line 45: “Emergent relationships between changes in remote sensing estimates of NPP and concomitant changes in ocean environmental conditions over the contemporary period can provide global constraints for Earth system models.”

- L50: Remove "similarly".

R3.10 We have removed the word similarly.

- L54–57: While I try to commend the efforts of fellow researchers as often as possible, I don't think this part of the manuscript is the right place for it. It is also unclear which part has been addressed by OC-CCI. I could be wrong, but my understanding is that OC-CCI merges all the "raw" satellite data (including light but also some derived products such as chlorophyll) but not NPP. If I'm correct, then OC-CCI addresses the issue of the time period and the data being used (the first and third items in the previous sentence), in which case it would be clearer to explicitly say so in the manuscript (otherwise the reader is left wondering what OC-CCI addresses). Hence, what about: "Sensitivity to the time period or the data being used has been recently addressed by the publication of a coherent multi-sensor satellite record spanning 1998–2023 that merges all available

single-sensor satellite missions with substantially reduced inter-sensor biases." (I would remove the following sentence: "The outcome is (...)".) This would also flow better logically with the following "Intrinsic differences in trends are however still expected from the range of algorithms available for quantifying NPP."

R3.11 We thank the reviewer for these suggestions and have amended the text as follows:

Line 50: "Trends in marine NPP estimated from remote sensing however also vary considerably depending on the time period, algorithm implemented, and data product being used¹²⁻¹⁵. Some of the sensitivities to time period and data product are addressed by the generation of a coherent multi-sensor satellite record spanning 1998-2023 that merges all available single-sensor satellite missions with substantially reduced inter-sensor biases³. Nonetheless, intrinsic differences in remote sensing trends are still apparent in the range of algorithms available for quantifying NPP rates."

- L59: Remove "that represent a range of different approaches to derive NPP" since this clear from the previous sentence.

R3.12 We thank the reviewer for this suggestion and have removed this sentence, please see the revised sentence below:

Line 56: "Here we focus on six algorithms including: (1) the 'vertically generalised production model's (Eppley-VGPM¹⁶ and Behrenfeld-VGPM¹⁷), which define phytoplankton growth as a function of chlorophyll-a, light and temperature, the difference being that Eppley-VGPM is an exponential function of temperature, while Behrenfeld-VGPM is a 4th order polynomial; (2) the 'carbon-based production models (Behrenfeld-CbPM¹⁸ and Westberry-CbPM¹⁹), which incorporate particulate backscatter as a proxy for phytoplankton carbon but differ in that Westberry-CbPM is both depth and wavelength resolved whilst Behrenfeld-CbPM is not; (3) the 'absorption-based production model' (Lee-AbPM²⁰), which defines NPP as a function of phytoplankton absorption rather than chlorophyll; and (4) the 'carbon, absorption, and fluorescence euphotic' resolving model (Silsbe-CAFE²¹), which integrates the learning from all the above algorithms to define NPP as a function of energy absorption and efficiency (for more details please see Methods)."

- L60–63: This list of 4 algorithms confused me at first because I was expecting 6 instead. I think it would be best if the 6 algorithms were defined here, which would avoid making the reader stumble on first read of "Lee-AbPM and Silsbe-CAFE" L68, since these are not defined at this stage in the manuscript. I would also recommend avoiding the single quotes here. E.g., what about: "These algorithms include two vertically generalised production models (Eppley-VGPM and Behrenfeld-VGPM), (and so on...)"

R3.13 We thank the reviewer for this suggestion and have amended the text as follows:

Line 56: "Here we focus on six algorithms including: (1) the 'vertically generalised production model's (Eppley-VGPM¹⁶ and Behrenfeld-VGPM¹⁷), which define phytoplankton growth as a function of chlorophyll-a, light and temperature, the difference being that Eppley-VGPM is an exponential function of

temperature, while Behrenfeld-VGPM is a 4th order polynomial; (2) the ‘carbon-based production models (Behrenfeld-CbPM¹⁸ and Westberry-CbPM¹⁹), which incorporate particulate backscatter as a proxy for phytoplankton carbon but differ in that Westberry-CbPM is both depth and wavelength resolved whilst Behrenfeld-CbPM is not; (3) the ‘absorption-based production model’ (Lee-AbPM²⁰), which defines NPP as a function of phytoplankton absorption rather than chlorophyll; and (4) the ‘carbon, absorption, and fluorescence euphotic’ resolving model (Silsbe-CAFE²¹), which integrates the learning from all the above algorithms to define NPP as a function of energy absorption and efficiency (for more details please see Methods).”

- L64–64: I would remove the obvious "Whilst each algorithm possesses different uncertainties and caveats for estimating NPP" and start the sentence with "None of algorithms has been found (...)" ("singular" is unnecessary and may be confusing).

R3.14 We thank the reviewer for this suggestion, but please note we have now moved all of the discussions around Round Robin exercises to a new section “Assessing the merits of the different remote sensing algorithms”. The revised sentence now reads:

Line 244: “During Primary Production Algorithm Round Robin exercises^{22–24} no single algorithm has been found to perform best at all times and locations. However, there is a general reduction in the root mean square difference between remote sensing NPP estimates and direct field measurements for the Lee-AbPM and Silsbe-CAFE algorithms (relative to the VGPM and CbPM algorithms), suggesting that they perform best overall (Supplementary Information Figure S9^{21,24–26}). Indeed, more recent studies that applied the Behrenfeld-VGPM, Westberry-CbPM and Lee-AbPM algorithms to OC-CCI data report similar findings where Lee-AbPM has the lowest RMSE (Wu et al., 2024).”

- L71–74: This sentence is a bit confusing and uses slightly imprecise language in my opinion. What about: "We ranked 15 CMIP6 Earth system models according to their ability to capture the emergent contemporary relationships between NPP and environmental variables (sea surface temperature, chlorophyll-a, and mixed layer depth) observed in the 6 remote-sensing algorithms." I think saying these relationships are "mechanistic" here was too much of a stretch, given these relationships are more akin to simple correlations. In addition, "parallel" is a little imprecise and the concomitance of the compared relationships can be delegated to the Methods section.

R3.15 We thank the reviewer for this suggestion and have amended the sentence as follows:

Line 70: “In this work, we rank fifteen CMIP6 Earth system models according to their ability to capture the emergent contemporary relationships observed between NPP and environmental variables (sea surface temperature, chlorophyll-*a* and the mixed layer depth) in the 6 remote sensing algorithms.”

- L74–78: This sentence is a bit convoluted and would probably read better if it started with the 4 rankings that "agree" (the word "bifurcation" is probably not the best here

either). What about: "Four algorithms (which includes the best performing algorithms according to XXX; Lee-AbPM and Silsbe-CAFE) concur that climate models projecting greater NPP declines rank higher, while the remaining two (Eppley-VGPM and Behrenfeld-CbPM) rank models that project slightly positive NPP trends higher." (about the "XXX" above: I would be explicit about what makes Lee-AbPM and Silsbe-CAFE better performers; I think the authors are referring to the round robin here, as they do L232, but I am not entirely sure. Please confirm)

R3.16 We thank the reviewer for this suggestion and have amended the sentence as follows:

Line 217: "Five algorithms concur that climate models projecting greater NPP declines rank higher, whilst the remaining algorithm (Eppley-VGPM) ranks models that project slightly positive NPP trends higher (Fig. 4)."

Please note that as we now have a new section where we assess the merits of each algorithm we no longer discuss which algorithms we think are the best. Instead this section immediately follows the ranking and similarly concludes that Lee-AbPM and Silsbe-CAFE are best algorithms.

- L78–80: This "assessment" sounds a little vague here. What about something more factual: "Furthermore, using the Lee-AbPM and Silsbe-CAFE algorithms also produce the most effective rankings (effectiveness is quantified by the reduction of inter-model variance when discarding lower ranking models)."

R3.17 We thank the reviewer for this suggestion; however please note that due to the revision of the manuscript this sentence has been deleted, and we no longer discuss inter-model variance.

- L80–81: NPP decline is always likely. What about something stronger (and that repeats the same language of "decline" rather than "loss"; repetition is good here): "These results suggest that future NPP decline is more likely than not, and this decline is currently underestimated by even the best ranked CMIP6 models, which predict the most intense NPP declines."

R3.18 We thank the reviewer for this suggestion have amended the text as follows:

Line 76: "These results suggest that future NPP decline is more likely than not, and this decline is currently underestimated by even the best ranked CMIP6 models, which predict the most intense NPP declines."

- L90 but also L93, L98, L108, L110, L113, the "S" before the Figure number is missing in "Supplementary Figure X".

R3.19 We thank the reviewer for finding this error and we have gone through the manuscript to ensure that all figures are labelled correctly in the text.

- L91 and throughout, in my opinion, there is a bit too much importance given to p-values versus the actual science or mechanism being discussed. For example, here in L91, the more important bit of information is that the increases in NPP are small. Maybe this is my personal aversion to statistical jargon, but I think that most of the p-value mentions should be relegated to Figure captions or supplementary Tables so that the main text is focused on the main message. Another issue I have is that I am not sure that I can formulate the null hypothesis that these p-values are based on in some (if not most) instances, which means that I am unable to truly interpret their meaning anyway (but, again, this could be just me).

R3.20 We thank the reviewer for this comment, but please note that upon revision of the manuscript we have altered the methodology of how we calculate the area-weighted mean-normalised NPP trends. We now use a jackknife resampling approach for each algorithm and report in the text the jackknife mean \pm stdev NPP trend. With this change in methodology we have changed the focus of the text to instead be on the trends themselves and the variance across jackknife simulations, with no mention of the statistical significance.

- L101–116: The statistical part of this paragraph is a bit confusing to me. Maybe it could be streamlined a little to emphasize the science instead of the statistical tests? It would also maybe be useful to move the last sentence up to the start of the paragraph.

R3.21 We thank the reviewer for raising this concern, we have since restructured the manuscript to move this discussion to a new section “Assessing the uncertainties of the different remote sensing algorithms”. Please see the revised text below:

Line 254: “In addition, the Jackknife trend analysis we conducted on the time series (Supplementary Information Fig. S10) demonstrates that both the Eppley-VGPM and Behrenfeld-VGPM algorithms are strongly sensitive to the start or end dates of the time series (Supplementary Information Fig. S10a-d), with high coefficients of variation and even a switch in the dominant direction of NPP trends across the simulations. Although both CbPM algorithms had similarly high coefficients of variation across the globe (relative to the VGPM algorithms), they remain dominated by negative trends across all simulations, with some evidence of an increase in the magnitude of negative trends and the number of positive trends in response to a change in the start and end dates (Supplementary Information Fig. S10e-h). The Lee-AbPM and Silsbe-CAFE algorithms displayed the most robust response in NPP trends to the jackknife simulations,

with much lower coefficients of variation and no tangible increase in the number of positive trends (with only a slight increase in the magnitude of negative trends, Supplementary Information Fig. S10i-l). Those areas of the globe that display relatively higher coefficients of variation (e.g. the Southern Ocean) thus represent regions with reduced confidence in the magnitude of the predominantly negative trends, but not in their direction. Overall, this indicates that there are larger uncertainties for global NPP trends from the VPGM and CbPM algorithms, relative to the trends estimated from Lee-AbPM and Silsbe-CAFE. Together these points of consideration around NPP algorithm validation and trend sensitivity to the jackknife simulations suggest that the Lee-AbPM and Silsbe-CAFE algorithms are the most robust and therefore best suited for the implementation of the model ranking scheme. Consequently, these results support a greater likelihood of global NPP declines into the future.”

- L118: What about "concomitant" in place of "parallel"?

R3.22 We thank the reviewer for this suggestion and have amended the text as follows:

Line 101: “Trends in NPP occur in response to concomitant modifications of the ocean environment that span ‘bottom up’ factors like resource limitation to ‘top down’ controls such as grazing.”

- L125–127: What about: "To statistically assess what locally drives changes in NPP, we use multiple linear regressions of contemporary trends in NPP against 4 environmental and biological drivers, for each remote-sensing algorithm."

R3.23 We thank the reviewer for this suggestion and have amended the text as follows:

Line 108: “To statistically assess what drives local trends in NPP, we use multiple linear regressions (MLR) that account for unequal variance and autocorrelation. We used MLRs to link contemporary trends in NPP to a suite of environmental and biological drivers across all algorithms and jackknife trend simulations (see Methods).”

- L127: I think it is important here to mention that warming SST can drive NPP in both directions. Increased stratification means less nutrient supply and thus NPP decline, while increases in metabolic rates are generally expected to increase NPP. One of the reasons I think this is important is because I have done a similar driver-decomposition exercise recently myself and I found that the compensation between warming (enhancing production) and the decline in nutrient supply was quite strong for my model (see Pasquier et al., 2024, Fig. 1, <https://bg.copernicus.org/articles/21/3373/2024>, but please note that I do not think the authors should cite me here)

R3.24 We thank the reviewer for this suggestion and for sharing with us this very interesting study, we have amended the text as follows:

Line 111: “These drivers are trends in annual mean sea surface temperature (SST; where warming increases phytoplankton metabolic rates and may retard nutrient supply due to greater ocean stratification), annual

mean chlorophyll-*a* concentration (CHL; which reflects phytoplankton biomass and physiology), and annual mean mixed layer depth (MLD; which impacts adjustments in both light and nutrient supply).”

- L132–142: What about something shorter, less detailed, and more to the point. For example, for the sentence starting L132, something like: "Using all four drivers significantly improved the multiple linear regressions for all remote-sensing algorithms." I would recommend keeping the gist of which remote-sensing algorithms had the most skillful regressions and move the statistical details (p values, R² values, and co) to the supplementary information.

R3.25 As per our prior responses, we do think it is important to retain the quantitative rigour alongside the narrative. Relegating all quantifications to the supplementary seems a bit extreme, but is ultimately a stylistic decision. We have removed some, but not all.

- L144–161: I think this paragraph on coefficients needs reworking. In particular, the main results must stand out and be placed upfront. In my opinion, the most important is that NPP is driven predominantly by SST, then CHL, then MLD. The second most important (which should therefore be discussed after the main point) are the spatial distributions and the mechanistic interpretations.

R3.26 We thank the reviewer for the stylistic suggestions, we have amended the structure to emphasise the key findings. This paragraph now begins:

Line 131: “The MLR coefficients associated with each driver show a reduction in amplitude, roughly halving in strength from SST to CHL and again from CHL to MLD (Fig. 3a-c). This indicates that trends in SST and CHL are the most important predictors of trends in NPP, whilst MLD plays only a minor role.”

- L165–167: I would not say that things "can be" done when things "have been" done. What about something like: "We apply the same multiple linear regression of NPP against SST, CHL, and MLDs to 15 CMIP6 Earth system models and rank these models according to their capacity to capture the emergent relationships observed with the remote-sensing algorithms and data. Specifically, we (...)"

R3.27 We thank the reviewer for this suggestion and have amended the text as follows:

Line 155: “Using an ensemble of fifteen Earth system models from CMIP6 we evaluate modelled trends in NPP (Fig. 1) in relation to the same set of drivers used in the remote sensing analysis to develop a ‘process based’ model ranking scheme.”

- L175–189: As for the similar paragraph on remote-sensing regressions, I would start with the most important point, which is that the coefficients are different in magnitude globally, and then move to the more detailed discussion of the distributions. The authors

should also consider discussing the mechanistic relationships that are explicitly built in these models, in the same way that Fig. S1 shows the built-in relationships of NPP with input variables for remote-sensing algorithms. In biogeochemistry models, NPP is explicitly related to temperature and chlorophyll as far as I know, and my intuition is that these relationships would heavily influence the regressions. I guess this might also help some interpretations.

R3.28 Again, we appreciate the advice to make the key results stand out better. We have adjusted the topic sentence.

Line 166: “Both the magnitudes and spatial distribution of the MLR coefficients across SST, CHL and MLD for each Earth system model reveal stark differences, relative to the remote sensing assessment (Fig. 3d-f). However, the general decline in their relative contribution to NPP trends from SST to CHL and lastly MLD largely remains, albeit to a lesser extent than the remote sensing algorithms.”

We also agree that the mechanistic relationships explicitly built into the different earth system models would strongly influence their regressions. We have tried to include some of this into the discussion by including the following sentence:

Line 127: “The higher global mean R^2 values for the VGPM algorithms is perhaps not surprising as the MLR is constructed using two of the three algorithm inputs, SST and CHL, with photosynthetically active radiation the remaining input variable.”

As an aside, NPP is modelled using the growth rate, light and nutrient limitations and the biomass standing stock in biogeochemical models used in the CMIP exercise. So there is not an explicit direct link to chlorophyll, except for an indirect linkage to the sensitivity to light limitation in some models.

- L194: I would recommend hand-holding here to explain what high/low EMD means, maybe simply a parenthesis with something like: "(low EMD means good agreement and thus high rank)" (but maybe it is the other way around, or maybe worse I misunderstood completely).

R3.29 We thank the reviewer for this suggestion and have added the following sentence to improve the clarity of the message:

Line 186: “This ranking is based on the dimensionless Earth mover’s distance (EMD) metric³², which quantifies the effort required to transform the distribution of the Earth system model MLR coefficients to match those obtained from each of the six remote sensing NPP algorithms. A low EMD value indicates that the Earth system model MLR coefficients closely match, i.e. are in good agreement, to those of the remote sensing algorithms.”

- L205: Add "Earth system" in "between remote sensing and Earth system models for these two variables"

R3.30 We thank the reviewer for this suggestion but please note that upon revision of the manuscript this sentence has since been deleted.

- L214: As much as I like short, clear, strong statements, I think this one is a bit too strong, and I think it is best to say which way Z scores improve ranking rather than the other way around. What about: "A low Z score thus indicates that the NPP–driver relationship in the Earth system model matches that of the remote-sensing algorithm well".

R3.31 We thank the reviewer for this suggestion and have amended the sentence:

Line 204: "The Z-score is defined as the distance of a value to the group mean, such that high Z-scores indicate values that are atypical and much larger than the mean and vice versa. A low Z score thus indicates that the NPP driver relationship in the Earth system model more closely matched that of the remote sensing algorithm."

- L215: Does "combine" here mean "sum"? If yes, I would suggest using "sum" and remove "using equal weighting".

R3.32 Combine here does not mean sum, but rather means average them together. We wanted to avoid the repetition of using the word 'mean' or 'average' in this section as we are combining the EMD mean and standard deviation values. We have however clarified what we did in parenthesis by specifically stating that we combined them by averaging them.

Line 207: "We then combine both Z-scores (from the EMD mean and standard deviation) using equal weighting (i.e. we averaged the Z-scores), before sorting the combined Z-scores from smallest to largest to rank each Earth system model's relative performance (Fig. 4)."

- L218: algorithms don't "manage" to reduce Δ NPP standard deviation. It would also help to reiterate what reducing across-model variance implies here. What about: "Only for the Eppley-VGPM, Behrenfeld-VGPM, Lee-AbPM, and Silsbe-CAFE algorithms does removing low-ranking Earth system models significantly reduce the across-model variance of Δ NPP, indicating more effective ranking (ref)." (I would then remove the sentence L225–227)

R3.33 We thank the reviewer for this comment but please note that we no longer include this statement any longer due to the change in our methodology.

- L227–231: I would rephrase this as something simpler like: "The Lee-AbPM and Silsbe-CAFE algorithms both produce the most effective rankings and rank Earth system models with negative future NPP predictions the highest." and remove "The remaining algorithms do not display any marked divergence in Δ EMD mean or standard deviation"

R3.34 We thank the reviewer for this comment but please note that we no longer include this statement any longer due to the change in our methodology.

- L235: Given the suggestion above that contains part of this sentence, I would rewrite as: "Together this suggests a greater likelihood of global NPP decline in the future."

R3.35 We thank the reviewer for this suggestion, however please note that with the revisions this statement has now been moved to "Assessing the merits of the different remote sensing algorithms". Please see the revised statement below:

Line 270: "Together these points of consideration around NPP algorithm validation and trend sensitivity to the jackknife simulations suggest that the Lee-AbPM and Silsbe-CAFE algorithms are the most robust and therefore best suited for the implementation of the model ranking scheme. Consequently, these results support a greater likelihood of global NPP declines into the future."

- L268: What about: "Remote sensing is a powerful tool".

R3.36 We thank the reviewer for this suggestion and have amended the sentence:

Line 306: "Remote sensing is a powerful tool for understanding changes in ocean properties over the contemporary period, with multi-decadal records commonly used to assess and constrain Earth system models' ability to accurately represent spatial and temporal variability in ocean processes."

- L273: There is one reference but this sentence mentions previous studies (plural). Maybe the authors meant to add more references here?

R3.37 We only provided one study as an example, hence it being preceded by e.g.. Furthermore this is one of the only studies we could find that specifically looked at ranking NPP in Earth system models, whilst other studies have focused on other processes. To keep this part concise and avoid confusion we chose to only reference the relevant study.

- L279: What is "the resource limitation diagnostics in Earth system models"? Is there a reference for it?

R3.38 What we were referring to as the resource limitation diagnostics in Earth system models are the penalties applied to phytoplankton growth in the model when a requirement for growth is at a suboptimal level. For example, models that contain the iron cycle will have an iron limitation term that lowers the modelled phytoplankton growth if the available concentration is below the estimated requirement. We have amended the text to provide some examples for clarity:

Line 316: “Future assessments should not only consider the uncertainties inherent to remote sensing algorithms (Supplementary Information Fig. S8), despite the complexity in deriving them (Song et al., 2024), but should also expand on the Round Robin intercomparison exercises (Supplementary Information Fig. S8) as more in situ data becomes available. Furthermore, future model assessments should consider using additional parameters in combination with those proposed here, such as the resource limitation diagnostics in Earth system models (e.g. iron limitation, light limitation etc), which could be used to assess ongoing changes in the Southern Ocean³³ and the equatorial Pacific³⁴.”

- L304: Is the code available publicly? (E.g., on a public repository such as GitHub, or better yet, in a public archive such as Zenodo.)

R3.39 We thank the reviewer for this suggestion and we now provide 2 resources. The first is JuPyTer notebook located here:

https://github.com/tjryankeogh/global_npp_trends, and the second is a Zenodo repository: <https://zenodo.org/records/14185537>. Any reader who should wish to replicate the figures and outcomes of the study can now do so with these resources. Additional, we have added the following statement to the data availability section:

“All data used in this study are available at <https://zenodo.org/records/14185537> and the code is available at https://github.com/tjryankeogh/global_npp_trends.”

- L362+ Methods section on MLR and EMD: I find this section quite hard to read with a number of occurrences of imprecise or convoluted wording. I think more equations and symbols here would help navigate the rather complicated assemblage of metrics. For example, among other things, I wonder if "normalized to the mean along the time dimension" means "normalized by the time-mean". I also wonder what checks and tests were conducted. I wonder what a significant pixel is. Equation 1 is not displayed correctly (I see a dotted square in the integral). "I1" on the left-hand-side of Equation 1 is not defined. Equation 1 also looks like it is missing a sentence to introduce it. The sentence just after Equation 1 starts with "Where" with an upper case "W" when it should be a lower case "w", right? By "proportion" I think the authors mean "area" but I am not entirely sure. The "A" in "A mean and standard deviation was calculated" is strange, as "mean" and "standard deviation" are well-defined. Statements like "x is the value of the model" is obscure (what model? the value of what?). I don't mean to be disparaging with the series of critiques above but I do think that the authors should

clarify this section so that any interested reader can understand the details of the methods employed and reproduce each step.

R3.40 We thank the reviewer for these suggestions and have amended the section to be clearer for the readers to understand.

“Annual means of NPP, SST, CHL and MLD were first jackknife resampled to 80% of the time series, representing 7 different possible simulations, and then mean-normalised, i.e. the time series was divided by its mean. Multiple ordinary least-squares linear regressions (MLR) were then performed using the Statsmodel package⁴⁶ using a Heteroskedasticity and Autocorrelation Consistent covariance estimator, where the time lags for autocorrelation were calculated following Newey & West (1994), defined in Equation 6:

$$\text{Equation 6: } \text{lag} = \left\lfloor 4 \times \left(\frac{T}{100} \right)^{2/9} \right\rfloor$$

where T is the length of the time series, which in this case is 26 years for remote sensing and 165 years for the Earth system models. No MLR was performed for either a remote sensing pixel or model grid point if any variable was missing data from any year of the time series or if the variance for any of the drivers was ~0. MLR coefficients for each prospective driver were then excluded from further analysis if either the remote sensing pixel or model grid point were not significant (p>0.05). Comparisons between the observational data products and model MLR coefficients were performed using the Earth mover’s distance (EMD) metric³², also known as the Wasserstein distance in mathematics⁴⁷ and Mallow’s distance in statistics⁴⁸, defined here in Equation 7:

$$\text{Equation 7: } l_1(u, v) = \int_{-\infty}^{+\infty} |u - v|$$

where l_1 is the first EMD, u and v are the respective distributions of the MLR coefficients from remote sensing and Earth system models and U and V are the respective cumulative distance functions of u and v. The MLR coefficient values for both the remote sensing and models were restricted using the interquartile range (IQR) fence test, $IQR \pm IQR \times 3$, to remove any extreme outliers. The EMDs were calculated on a per biome basis using the biome classification of Fay & McKinley²⁷, with the EMD weighted by the biome's proportion (%) of the global ocean. The EMDs for SST, CHL and MLD were then averaged to generate an EMD mean and standard deviation per Earth system model. To rank the models the Z-score, also known as standard score, was calculated using Equation 8:

$$\text{Equation 8: } z = \frac{x - \mu}{\sigma}$$

where x is either the model’s EMD mean (or standard deviation), μ is the model ensemble mean of either the EMD mean (or standard deviation) and σ is the model ensemble standard deviation of either the EMD mean (or standard deviation.) The final Z-scores, determined from both the EMD mean and the EMD standard deviation, were then generated by combining with equal weighting (i.e. the Z-scores were averaged together).”

- Fig 1:

- While I understand that the authors computed "decadal trends of annual means", this sounds equivalent to simply "mean decadal trends".
- What is the normalization used for NPP trends?

R3.41 We thank the reviewer for this suggestion and have amended the figure caption, but please also take note that we have amended how the data is represented in Figure 1.

“Figure 1: Variability of net primary production trends from CMIP6 Earth system models. (a) Area-weighted mean-normalised net primary production (NPP) decadal mean trends (% year⁻¹) calculated using ordinary least squares for the historical (1850-2014), contemporary (1998-2023) and future (2015-2100) periods for the CMIP6 Earth system model ensemble. (b) Area weighted ΔNPP (Pg C year⁻¹), calculated as the difference between the end of the historical period (1995-2014) and the end of the century (2081-2100), for each of the Earth system models in the CMIP6 ensemble. Both panels are sorted by ΔNPP from low to high values.”

- Fig 4: This figure is central to the manuscript, yet I think it could be improved a fair amount. I understand the intent of the authors to visualize the ΔNPP along the rankings, but these bar plots are all redundant with Fig 1. In addition, I simply find this Figure

painful to grasp at a glance, as it forces the reader to keep looking back and forth at the legend and to squint to distinguish colors. Furthermore, I think that the rankings themselves are a little misleading, in the sense that it does not show the Z score. As a solution to these issues, I would consider merging Fig 1 and 4 in the following way: First, sort the Earth system models by Δ NPP instead of alphabetically in Fig 1. (This is to prepare the merge with Fig 4 but it will also help with spotting the disagreements between NPP trends and Δ NPP.) Then, append a 3rd panel (panel c) at the bottom containing a heatmap (see, e.g., https://matplotlib.org/stable/gallery/images_contours_and_fields/image_annotated_heatmap.html) of the Z-scores (align the columns with the Earth system models of panels a and b, and use the rows for remote-sensing algorithms, also sorted by NPP trend.) By choosing a colormap for the heatmap that highlights the models that rank best, this will show at a glance the central message of the paper, add extra useful information visually (the Z scores), all while removing 1 Figure with 6 redundant panels. It will also place the central message in the first Figure, which is nice on the readers that get tired quickly. If the authors do follow this suggestion, they should make sure that the sorting of Earth system models is applied to all Figures to avoid confusion.

R3.42 We appreciate the constructive criticisms of the figures and have tried to take them all on board. Indeed, in doing so we reflected heavily on the order of presentation and the links to the underlying scientific messages. In short, the amended figures are:

“Figure 1: Variability of net primary production trends from CMIP6 Earth system models. (a) Area-weighted mean-normalised net primary production (NPP) decadal mean trends (% year⁻¹) calculated using ordinary least squares for the historical (1850-2014), contemporary (1998-2023) and future (2015-2100) periods for the CMIP6 Earth system model ensemble. (b) Area weighted Δ NPP (Pg C year⁻¹), calculated as the difference between the end of the historical period (1995-2014) and the end of the century (2081-2100), for each of the Earth system models in the CMIP6 ensemble. Both panels are sorted by Δ NPP from low to high values.”

“Figure 4: Ranking Earth system models using Z-score assessments of the Earth mover’s distance metric. Bar plots of mean \pm standard deviation Jackknife resampled ranked Earth system model Δ NPP (Pg C year⁻¹) for (a) Eppley-VGPM, (b) Behrenfeld-VGPM, (c) Behrenfeld-CbPM, (d) Westberry-CbPM, (e) Lee-AbPM and (f) Silsbe-CAFE NPP algorithms. All bars are coloured by the mean Z-score across the jackknife resampling exercise. Please note that the absence of an errorbar is indicative of the same model being ranked in the same position for all 7 of the jackknife simulations.”

“Figure S7: Ranking Earth system models using Z-score assessments of the Earth mover’s distance metric for each Jackknife simulation. Heatmaps of Z-scores for ranked Earth system models per remote sensing NPP algorithm, including (a) Eppley-VGPM, (b) Behrenfeld-VGPM, (c) Behrenfeld-CbPM, (d) Westberry-CbPM, (e) Lee-AbPM and (f) Silsbe-CAFE.”

- Fig S1: y-axis label mentions "normalized NPP". What this normalization is should be explained in the caption.

R3.43 We thank the reviewer for this suggestion and have amended the figure caption to state the normalisation scheme, max-normalised. Please note however that following a reviewer comment the previous Figure S1 has now become Figure S2.

“Figure S2: Exploring the input variable dependency in estimating net primary production. Line plots of max-normalised net primary production (NPP) calculated using the (a) Eppley-VGPM, (b) Behrenfeld-VGPM, (c) Behrenfeld-CbPM, (d) Westberry-CbPM, (e) Lee-AbPM and (f) Silsbe-CAFE NPP algorithms. Input variables include sea surface temperature (SST), chlorophyll-a (CHL), photosynthetically active radiation (PAR), particulate backscattering (b_{bp}), mixed layer depth (MLD), diffuse attenuation coefficient (K_d), phytoplankton absorption (a_{ph}) and detrital absorption (a_{dg}). The input variable being tested was allowed to range between the climatological (1998-2023) 20th and 80th percentile, whilst the other input variables were held constant at the climatological median value.”

- All the other figures are beautiful.

Thank you - your positive and encouraging inputs have been very constructive and appreciated.